# How Much Information Fits in a Vector?

**Christopher Gadzinski**                                       *christopher.gadzinski@uni.lu*
*Department of Computer Science*
*University of Luxembourg*

**Decebal Constantin Mocanu**                                  *decebal.mocanu@uni.lu*
*Department of Computer Science*
*University of Luxembourg*

**Reviewed on OpenReview:** *https://openreview.net/forum?id=Nby4pCPIZI*

## Abstract

Recent work in neural network interpretability has suggested that hidden activations of some deep models can be viewed as linear projections of much higher-dimensional vectors of sparse latent "features." In general, this kind of representation is known as a superposition code. This work presents an information-theoretic account of superposition codes in a setting applicable to interpretability. We show that when the number $k$ of active features is very small compared to the number $N$ of total features, simple inference methods currently used by sparse autoencoders can reliably decode a $d$-dimensional superposition code when $d$ is a constant factor greater than the Shannon limit. Specifically, when $\ln k / \ln N \leq \eta < 1$ and $H$ is the entropy of the latent vector in bits, we prove necessary and sufficient asymptotic conditions of the form $d/H > C(\eta)$ for certain constants $C(\eta)$. However, for a fixed $\eta$, we show empirically that the critical value of $d/H$ depends significantly on what decoding method is used. For example, when $\eta = 0.3$, an estimation procedure based on the popular top-$k$ activation function requires $d/H > 4$ dimensions per bit, while a method that requires only around 3 times the computational expense succeeds with $d/H > 2$. We hope this work helps connect research in interpretability with perspectives from compressive sensing and information theory.

## 1 Introduction

If each neuron in a given neural network coded for a "meaningful feature" of its input, we could hope to reverse-engineer the network's behavior on a neuron-by-neuron basis. However, many large models have been found to learn neurons that correlate simultaneously with apparently unrelated features. This phenomenon is known as polysemanticity (Nguyen et al., 2016; Zhang & Wang, 2023; Olah et al., 2020).

The appearance of so-called "polysemantic neurons" is not surprising from a connectionist viewpoint. Since at least the 1980s, proponents of artificial neural networks have argued that these systems may naturally use **distributed representations**—coding schemes where individual features are represented by patterns spread over many units of computation, and conversely where each unit carries information on many features. This term was apparently coined in Rumelhart et al. (1986), Chapter 3. In contrast, a *local* representation would dedicate each unit to a single feature. See Thorpe (1989) for a general discussion of local and distributed codes. Figure 1 illustrates a classic example of a coarse code, one kind of distributed representation.

As of yet, relatively little is known about how large neural networks learn to represent information in their hidden layers or to what extent this information can be interpreted. However, should "interpretable features" exist, the connectionist viewpoint makes it natural that they would be stored with non-local codes. This is a common assumption in interpretability research today; for example, when Meng et al. (2022) intervened on an MLP layer of a language model to "edit" a factual association, both the "subject" and the "fact" were modeled as vectors of activations rather than individual neurons.

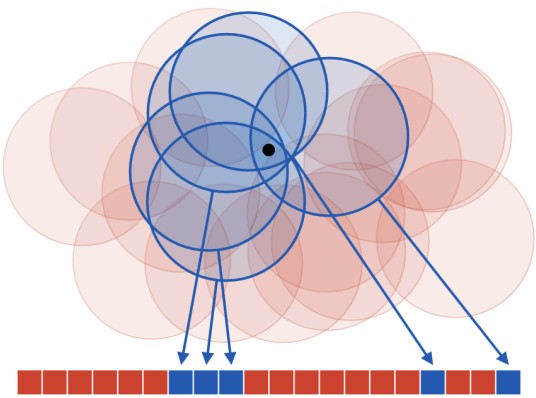

Figure 1: A coarse code representing a point on a plane. Each "neuron," drawn as a red or blue square, encodes whether the point belongs to an associated "receptive field." Although no neuron gives specific information on the position of the point, the overall code determines its position with reasonable accuracy.

To infer some kind of latent features from an activation vector $y$, one simple proposal is to model $y$ as a linear projection $y = Fx$ of a high-dimensional and *sparse* vector $x$. This idea is based on the remarkable fact that, for certain families of matrices $F$ and for certain constraints on the sparsity of $x$, a $d$-dimensional linear image $y$ codes uniquely for an $N$-dimensional sparse vector $x$ even when $N = \Omega(e^d)$.

We refer to the columns of $F$ as codewords and the whole matrix $F$ as a dictionary. Since $y$ is a linear superposition of codewords, it is called a **superposition code** for $x$. The task of inferring $x$ from a superposition code is known as sparse reconstruction, and the task of inferring the dictionary $F$ from a distribution over the codes $y$ is called dictionary learning. Both of these problems have been studied in the field of compressive sensing, although with different applications in mind; see Elad (2010) for a review of classic work in the context of signal and image processing.

Already in 2015, Faruqui et al. (2015) used a dictionary learning method to derive sparse representations for word embeddings and argued that these latents were more interpretable than the original embedding dimensions. More recently, a series of works beginning with Yun et al. (2021) have applied dictionary learning to the internal representations of transformer-based language models. Cunningham et al. (2023) suggested the use of **sparse autoencoders** (SAEs) and Templeton et al. (2024); Gao et al. (2024) scaled sparse autoencoders to production-scale large language models.

Templeton et al. (2024) showed that latent features learned by SAEs are often highly interpretable, and that intervention on these features allows "steering" language models in predictable ways. However, as reported in Gao et al. (2024), even SAEs with extremely large numbers of latents suffer from an apparently irreducible reconstruction error. According to Sharkey et al. (2025), understanding the limitations of SAEs—and dictionary learning in general—is an important open question in the research program of mechanistic interpretability.

## 1.1 Contributions

To infer a latent representation $x \in \mathbb{R}^N$ from an activation vector $y \in \mathbb{R}^d$, sparse autoencoders use a simple estimate of the form $\hat{x}(y) = \sigma(Gy)$ where $G \in \mathbb{R}^{N \times d}$ is a learnable matrix and $\sigma$ is some kind of thresholding operation. Throughout this paper, we refer to any estimate of this form as a "one-step estimate" for $x$, since it is not an iterative method. Gao et al. (2024), a work representative of research on large-scale sparse autoencoders, considered latent vectors with dimension $N$ on the order of $2^{20}$ and with $k \approx 64$ non-zero coefficients each.

Research on sparse autoencoders is mired in many kinds of scientific uncertainty. Of course, it is not known a priori how well activation vectors can be effectively modeled as sparse superposition codes. It is not even necessarily clear what is meant by an "interpretable feature," as per Zytek et al. (2022). Furthermore, even if some activity of a model can be modeled as a superposition code, it is not obvious whether the latent features being encoded can be recovered by a one-step estimate. Indeed, the compressive sensing literature provides

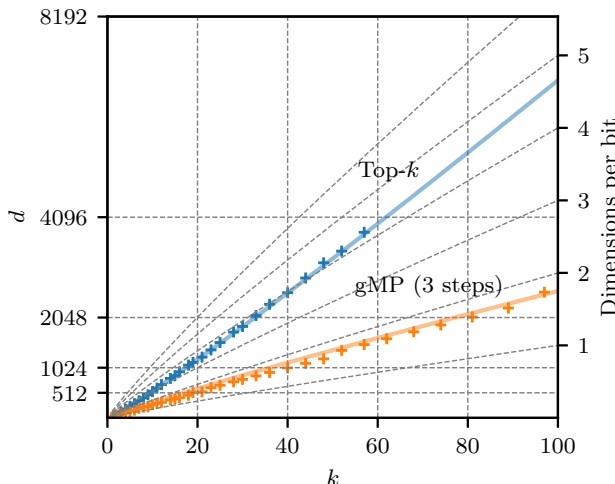

Figure 2: Crosses give minimum embedding dimensions $d$ at which two different methods were empirically found to completely decode a superposition code for a $k$-sparse subset of $\{1, \ldots, 2^{20}\}$ with probability 0.95, and colored lines give rules of thumb derived from our theoretical discussion in Section 4. The inverse "bitrate" $d/\tilde{H}_2$, where $\tilde{H}_2 = k \log_2(eN/k) \approx \log_2\binom{N}{k}$, is indicated by the right axis. Top-$k$ is a method currently used by sparse autoencoders, and generalized matching pursuit (gMP) is a simple algorithm described in Section 2.2.

many iterative methods for sparse recovery that require more computation but succeed more reliably. To what extent are the limitations of SAEs explained by the limitations of our sparse recovery methods?

One way to frame this question is to ask *how much information* a superposition code may hold. Classically, this is addressed by the tools of information and coding theory. Given a "channel" with certain characteristics—for example, a band-limited telephone connection or a binary storage device with a certain failure rate—the amount of information we can encode is asymptotically characterized by a channel "capacity" measured in bits per unit of channel usage.

The goal of this paper is to perform a similar analysis for sparse superposition codes in a setting that is meaningful to sparse autoencoders. More specifically, we consider a situation where the latent vector $x$ is extremely sparse and must be recoverable from the code $y$ by a relatively efficient method. We will focus on a specific toy example described in Section 2. Our main contributions are the following.

1. Asymptotically in a regime of sublinear sparsity (meaning that $k \le N^\eta$ for some $\eta < 1$), we provide theoretical results on the number of dimensions $d$ required for certain one-step methods to reliably decode superposition codes. Our conclusions can be summarized in terms of the ratio $d/H$, where $H$ is the entropy of the sparse latent vector (Corollary 2). Empirically, we find that our results give an accurate rule of thumb for empirical performance of these methods when $\eta \in (0, 0.4)$, even as $N$ ranges from $2^8$ to $2^{20}$.

2. We investigate a simple extension of top-$k$ which we call **generalized matching pursuit** (gMP). By numerical experiment, we show that gMP can decode superposition codes carrying more than two times as much information while requiring, for the same values of $(N, d, k)$, only a small constant factor more computation than top-$k$.

A concrete result of our findings is illustrated in Figure 2. For $k \in \{1, \ldots, 100\}$, we show the number of dimensions $d$ required for two different methods to accurately read a $k$-sparse subset of $\{1, \ldots, 2^{20}\}$ that has been transmitted as a $d$-dimensional superposition. (The dictionary $F$ used is a random Rademacher dictionary; see Section 2 for more details.) As $k$ increases, the ratio $d/H$ of required dimensions per bit of entropy increases when we use a simple decoding method based on the top-$k$ activation function. For moderate values of $k$, this method requires more than 4 dimensions per bit. In particular, a superposition code with more than 60 elements requires an embedding dimension greater than 4096. In contrast, generalized matching pursuit (gMP) with $T = 3$ steps requires less than two dimensions per bit, while requiring only around 3 times as many FLOPs for the same values of $(N, d, k)$.

## 1.2 Relation to Other Work

The basic problem of recovering a sparse vector $x$ from a linear projection is studied in compressive sensing. This field stems from works such as Donoho (2006) and Candes & Tao (2005), which showed that systems of linear measurements can be used to reconstruct certain kinds of sparse signals even when the number $d$ of linear measurements is much smaller than the nominal dimension $N$ of the signal. This is an important observation in signal processing, since many real-world signals are known to admit sparse representations in some (potentially redundant) basis. For example, images often admit sparse approximations in a wavelet basis; see Chapter 10 of Elad (2010).

A sparse signal can naturally be encoded (either exactly or approximately) with only a small fraction of the information that would be required to store each of its $N$ dimensions explicitly. The surprising result of compressive sensing is that compression can be performed by a system of *non-adaptive* linear measurements— that is, measurements (or "sensors") which are fixed a priori—in such a way that decompression can be performed efficiently. This is the explanation for the term compressive (or compressed) sensing. On the other hand, the subsequent operation needed to reconstruct the sparse signal $x$ from the vector of linear measurements $y$ is necessarily non-linear. Two important theoretical models for this operation are the solution of the linear program

$$\textbf{minimize } \|x\|_1 \quad \textbf{subject to } Fx = y$$

and the algorithm of approximate message passing (Donoho et al., 2009).

In compressive sensing, the matrix $F \in \mathbb{R}^{d \times N}$ is often called the design matrix. For linear projections $Fx$ to faithfully encode sparse vectors $x$, it is intuitive that the rows of the design matrix must not be too aligned with the basis vectors of $\mathbb{R}^N$. More precisely, the *nullspace property* (Cohen et al., 2009) formalizes the idea that the nullspace of $F$ must not contain elements which are "too sparse." Remarkably, it can be shown that design matrices with independent and randomly sampled entries have good compressive sensing properties.

The idea of using measurements with randomly chosen coefficients bears a resemblance to the idea that, in a distributed representation, each unit of storage should integrate over many different "features" of the information being stored. (Such measurements are also called *holographic*, and are reminiscent of the operation of optical holography.) Connections between compressive sensing and vector representations used in deep learning have been identified before, for example in Arora et al. (2015).

Compressive sensing is a deep field of study and has received considerable attention over the last two decades. Given a model for the underlying signal and the system of measurements, a typical problem is to characterize the minimum number of measurements $d$ required for a certain reconstruction method to achieve some standard of performance. This minimal value is called a sample complexity. For example, Reeves et al. (2019) studied the sample complexity of the maximum likelihood estimator in a regime of sublinear sparsity and in the presence of a small amount of measurement noise. Many practical methods for sparse reconstruction are available, and more continue to be developed; for example, Takeuchi (2024) extended the approximate message passing algorithm to achieve better sample complexity for signals with sublinear sparsity.

However, many results of compressive sensing are not immediately relevant to the study of sparse autoencoders in interpretability. First, in compressive sensing it is common to consider a regime of *linear sparsity*, where $k \geq \epsilon N$ for some $\epsilon \gg 0$, whereas in latent representations found by SAEs the ratio $k/N$ is typically extremely small. Furthermore, even in works that consider sublinear sparsity, decoding methods are typically either iterative (Truong, 2023; Takeuchi, 2024) or require specially designed dictionaries (Li et al., 2019). Although prior works like Bajwa et al. (2010) have studied "one-step" methods for sparse reconstruction, we were not able to find a comparison of one-step methods and iterative methods in a regime meaningful for SAEs. The present work serves as an analytical starting point on this topic. Further details on the relation of our results to compressive sensing are discussed in Section 4.3.

### 1.3 Structure of This Work

The remainder of this paper is structured as follows.

- Section 2 introduces the toy superposition code to be studied and the decoding methods we will consider. These are MAP (maximum a posteriori) threshold decoding, top-$k$ decoding, and gMP (generalized matching pursuit).

- Section 3 briefly recalls the idea of a Shannon limit and explains some heuristic predictions for sample complexity.

- Section 4 gives our theoretical results on the performance of threshold and top-$k$ decoding as well as our numerical comparison with gMP.

- Finally, Section 5 discusses our choice to use random dictionaries.

## 2 Superposition Codes and Sparse Recovery

We begin by describing the toy scenario to be studied. Given a large number $N$, consider a map $F$ that represents each subset $x \subseteq [N] = \{1, \ldots, N\}$ by a linear combination

$$y = Fx = \sum_{i \in x} f_i \in \mathbb{R}^d,$$

where the vectors $\{f_i \in \mathbb{R}^d : i \in [N]\}$ are chosen in advance and where the dimension $d$ of the encoding is expected to be much smaller than $N$. The vectors $f_i$ are called codewords, the collection $\{f_1, \ldots, f_N\}$ is called a dictionary, and the image $Fx$ is called a superposition code. By viewing $x$ as a vector in $\{0, 1\}^N$ with coefficients

$$x_i = \begin{cases} 1 \colon i \in x \\ 0 \colon \text{otherwise} \end{cases}$$

and viewing $F$ as the matrix of column vectors $[f_1 \ \ldots \ f_N]$, we can write $y = Fx$ as a linear equation. In this work, we'll model our subset as a random variable $X$ uniformly distributed over all subsets of some fixed size $k$. In keeping with the orders of magnitude discussed in Gao et al. (2024), we are interested in values like $N \approx 2^{20}$ and $k \approx 64$; in particular, we assume $N \gg k$.

In general, the problem of recovering a sparse signal $x$ from a linear projection $Fx$ is known as sparse recovery. In this work, we will use language from coding theory and think of $x$ as some data to be encoded and the projection $y = Fx$ as a code. Thus, sparse recovery becomes the task of "decoding" $x$. Note that this is opposite to the convention used in sparse autoencoders, where the map estimating $x$ from $y$ is called the "encoder." (In classical applications of autoencoders, the learned representation is typically understood as an efficient code for the data distribution and has fewer dimensions than the data being modeled.)

Due to a connection with sparse linear regression, the problem of identifying the support of a vector $x$ from a linear projection $Fx$ is often called model selection. When the coordinates of $x$ take values in $\{0, 1\}$, sparse recovery and model selection coincide. In general, note that model selection presents the main difficulty of sparse recovery; once the support of $x$ is identified, its non-zero coordinates can be estimated easily, e.g. by solving a least squares problem.

In practice, a dictionary $F$ employed by a neural network will likely have special properties related to the nature of the data being encoded and the operations that need to be performed. However, for the purposes of encoding a uniformly random subset of $[N]$, one natural way to choose a dictionary is to make each codeword an independent random vector. The simplest motivation for this strategy is the fact that, so long as $d = \Omega(\ln N)$, a dictionary of $N$ randomly chosen vectors is highly "incoherent." For example, we have the following.

**Proposition 1.** *For $\mu > 0$ and $p > 0$, let $d \geq 2\mu^{-2}(2\ln N + \ln p^{-1})$, and let $\{F_1, \ldots, F_N\}$ be independent draws from $\{-1/\sqrt{d}, 1/\sqrt{d}\}^d$. Then*

$$\forall i \neq j, \; |\langle F_i, F_j \rangle| < \mu \tag{1}$$

*with probability at least $(1 - p)$.*

Throughout this work, our convention will be to write random variables (like the codewords $F_i$ above) with capital letters. For a standard proof of Proposition 1, see Appendix D.

The infimum of all constants $\mu$ satisfying the condition (1) is called the mutual coherence of the dictionary $F$. Bounds on mutual coherence give a relatively simple way to guarantee the success of various sparse recovery algorithms; for example, see Chapter 4 of Elad (2010). In our scenario, we can say the following.

**Proposition 2.** *Given a dictionary $F = \{F_1, \ldots, F_N\} \subseteq \mathbb{R}^d$ of unit norm codewords with mutual coherence smaller than $\frac{1}{2k}$, for any $k$-sparse vector $x \in \{0,1\}^N$ and for all $i = 1, \ldots, N$ we have*

$$x_i = \begin{cases} 1 \colon \langle F_i, Fx \rangle \geq 1/2 \\ 0 \colon \textit{otherwise.} \end{cases}$$

*Proof.* Suppose that $x_i = 0$ for a certain index $i$. By assumption that the mutual coherence of $F$ is smaller than $\frac{1}{2k}$, we have $\langle F_i, F_j \rangle < \frac{1}{2k}$ for all $i \neq j$, and so

$$\langle F_i, Fx \rangle = \sum_{x_j = 1} \langle F_i, F_j \rangle < \frac{k}{2k} = \frac{1}{2}.$$

Similarly, using that $\langle F_i, F_i \rangle = 1$ and $\langle F_i, F_j \rangle > -\frac{1}{2k}$ for all $i \neq j$, when $x_i = 1$ we find that

$$\langle F_i, Fx \rangle = \langle F_i, F_i \rangle + \sum_{x_j = 1 \wedge j \neq i} \langle F_i, F_j \rangle > 1 - \frac{(k-1)}{2k} > \frac{1}{2}.$$

Thus, all coordinates $x_i$ are guaranteed to be recovered from $Fx$ by thresholding the inner product $\langle F_i, FX \rangle$. $\square$

Overall we can already guarantee that, for a fixed value of $k$, the dimension $d$ of our superposition code need only grow logarithmically with respect to the size $N$ of the dictionary for a very simple method to succeed at recovering $x$ from $Fx$. More specifically, applying Proposition 1 with $\mu = \frac{1}{2k}$ and combining with Proposition 2 gives the following.

**Corollary 1.** *Let $k \geq 1, N \geq 1$ be any integers, let $p > 0$, and suppose*

$$d \geq 8k^2(2\ln N + \ln p^{-1}).$$

*Then given a dictionary $F = \{F_1, \ldots, F_N\}$ in the conditions of Proposition 1, it is true with probability at least $(1 - p)$ that every $k$-sparse vector $x \in \{0,1\}^N$ can be recovered from $Fx$ by the formula*

$$x_i = \begin{cases} 1 \colon \langle F_i, Fx \rangle \geq 1/2 \\ 0 \colon \textit{otherwise.} \end{cases}$$

In the remainder of this work, we are interested in refining this prediction and understanding how sample complexity (the requirement on $d$) depends on the method we use to infer $x$. For simplicity we will consider only Rademacher dictionaries in our theoretical results, but in numerical experiments (Section 4.2) we find that dictionaries with codewords drawn uniformly from the unit sphere behave very similarly.

Finally, it remains to define the decoding methods we will study. We informally refer to methods that require only a single "step" of estimation and thresholding as one-step decoders. In this work, we will consider two one-step methods and one very simple iterative algorithm which we call generalized matching pursuit (gMP).

## 2.1 Threshold Decoding and Top-k Decoding

To motivate one-step decoders, consider the problem of estimating a single coefficient $X_i$ from the model

$$Y = X_i f_i + \overbrace{\sum_{j \neq i} X_j f_j}^{Z_i}. \tag{2}$$

If we replace the sum $Z_i$ with a centered Gaussian vector $\tilde{Z}$ of non-singular covariance $\Sigma$, the inference problem for $X_i$ becomes tractable; specifically, if we define an inner product by $\langle v, w \rangle_\Sigma = v^T \Sigma^{-1} w$, then the maximum likelihood estimator for $X_i$ is given by the linear function

$$\lambda_i(Y) = \frac{\langle f_i, Y \rangle_\Sigma}{\|f_i\|_\Sigma^2}.$$

In signal processing, estimating a scalar in the presence of noise is called filtering. In the context of a Gaussian model, estimates like $\lambda_i(Y)$ are called matched filters. In terms of our matched filter, we can express the exact posterior under the simplified model $Y = X_i f_i + \tilde{Z}$ as

$$\ln \mathrm{P}(X_i = x \mid Y = y) = C + \ln \mathrm{P}(X_i = x) - \frac{\rho}{2} \left( x - \lambda_i(y) \right)^2 \tag{3}$$

where $C$ is a normalizing constant and $\rho = (\mathrm{Var}\, \lambda_i(\tilde{Z}))^{-1}$ is called the signal-to-noise ratio (SNR). (See Appendix A for a full derivation.)

When $F$ is a dictionary of codewords drawn independently from an isotropic distribution and $X$ is independent from $F$, the vector $Z_i$ is isotropic. Thus, it is natural to take $\tilde{Z}$ to be an isotropic Gaussian. If $\|f_i\| = 1$, the maximum likelihood estimate for $X_i$ is then simply the inner product $\lambda_i(Y) = \langle f_i, Y \rangle$. Furthermore, in view of Equation (3), the maximum a posteriori (MAP) estimate for $X_i$ has the form

$$\hat{X}_i(Y) = \begin{cases} 1 \colon \lambda_i(Y) \geq \tau \\ 0 \colon \text{otherwise,} \end{cases} \tag{4}$$

where the threshold $\tau$ is given by

$$\tau = \frac{1}{2} - \frac{1}{\rho} \ln \frac{\mathrm{P}(X_i = 1)}{\mathrm{P}(X_i = 0)}.$$

Developing this expression and ignoring terms which will be negligible in practice yields

$$\tau_{\mathrm{MAP}} = \frac{1}{2} + \left( 1 - \frac{\ln k}{\ln N} \right) \frac{k \ln N}{d}.$$

(See Appendix B for the full derivation.) In the sequel, **threshold decoding** at level $\tau$ will refer to the estimation of the coefficients of $X$ by Equation (4) using the given value of $\tau$, and **MAP decoding** will refer to threshold decoding with $\tau = \tau_{\mathrm{MAP}}$.

Another natural strategy is to select the $k$ codewords whose corresponding estimates $\lambda_i(Y) = \langle f_i, Y \rangle$ are largest. We will call this **top-$k$ decoding** (Algorithm 2). In case of ties, top-$k$ is understood to choose exactly $k$ indices in an arbitrary way.

Top-$k$ has the disadvantage that $k$ must be known exactly in advance, whereas threshold decoding only requires $k$ to adjust the threshold $\tau$. However, should $k$ be known, it is clear that top-$k$ is more reliable than threshold decoding; that is, if threshold decoding succeeds at any level $\tau$, top-$k$ must also succeed.

Note that in some works (Elad, 2010) top-$k$ is known as the thresholding algorithm. Finally, note that in practice the latent quantities $X_i$ to be estimated generally take values besides 0 and 1. The two algorithms described here can be viewed as simplifications of one-step methods currently used by autoencoders. Sparse autoencoders using top-$k$ activations were first suggested by Makhzani & Frey (2014).

**Input:** $y \in \mathbb{R}^d$, $F \in \mathbb{R}^{d \times N}$, $\tau \in \mathbb{R}$
**Output:** $\hat{x} \in \{0, 1\}^N$
**1** $\hat{x} \leftarrow \mathbf{0}$ ;
**2** $\hat{x} \leftarrow 1$ where $F^T y \geq \tau$ ;
**3 return** $\hat{x}$

**Algorithm 1:** Threshold decoding

**Input:** $y \in \mathbb{R}^d$, $F \in \mathbb{R}^{d \times N}$, $k \in \mathbb{N}$
**Output:** $\hat{x} \in \{0, 1\}^N$
**1** $\hat{x} \leftarrow \mathbf{0}$ ;
**2** $\hat{x} \leftarrow 1$ at $k$ largest values of $F^T y$ ;
**3 return** $\hat{x}$

**Algorithm 2:** Top-$k$ decoding

## 2.2 Generalized Matching Pursuit

As discussed earlier, the field of compressive sensing provides various iterative algorithms for sparse recovery. For example, one classic starting point is to apply proximal gradient descent to the objective

$$\frac{1}{2d}\|Fx - y\|_2^2 + \frac{\lambda}{d}\|x\|_1$$

where the parameter $\lambda > 0$ can be understood as a Lagrange multiplier. This gives the iterative soft thresholding algorithm (ISTA) (Daubechies et al., 2003), and is the basis for methods like FISTA (Beck & Teboulle, 2009). A single iteration of ISTA involves a matrix-vector product followed by a thresholding operation, and so is computationally comparable to top-$k$ or MAP thresholding. Unrolling ISTA and optimizing its parameters, as we would do for a deep neural network, is called learned ISTA (LISTA) (Gregor & LeCun, 2010). Another notable family of methods is based on the approximate message passing algorithm (Maleki, 2010). However, for sparse autoencoders, a training run using top-$k$ is already very computationally expensive. Therefore, it is interesting to study the properties of relatively cheap modifications to top-$k$.

In this work, we suggest a simple iterative generalization of top-$k$, which we call **generalized matching pursuit** (gMP). Rather than finding the $k$ codewords that are most correlated with the superposition $y$ all at once, gMP iteratively selects sets of $k_t \leq k$ maximally correlated codewords and subtracts their sum from the value of $y$ supplied to the next iteration. We fix the number $T$ of iterations in advance, as well as the numbers $(k_1, \ldots, k_T)$ of codewords to be selected at each iteration. We suggest choosing the latter as evenly as possible subject to the constraint $k_1 + \cdots + k_T = k$. Explicitly, gMP is the following algorithm.

**Input:** Code $y \in \mathbb{R}^d$, dictionary $F \in \mathbb{R}^{d \times N}$, sparsity $k$, steps $T$
**Output:** Latent $\hat{x} \in \{0, 1\}^N$
**1** Choose any step sizes $k_1, \ldots, k_T$ such that $\sum_t k_t = k$ and $\max_t k_t$ is minimized ;
**2** $\hat{x} \leftarrow \mathbf{0}$ ;
**3 for** $t = 1, \ldots, T$ **do**
**4** $\quad r \leftarrow y - F\hat{x}$ ;
**5** $\quad \hat{x}_i \leftarrow 1$ at $k_t$ indices $i$ satisfying $\hat{x}_i = 0$ and maximizing $(F^T r)_i$ ;
**6 end**
**7 return** $\hat{x}$

**Algorithm 3:** Generalized matching pursuit (gMP) with $T$ steps

Since the most computationally expensive step of Algorithm 3 is to compute the product $F^T r$, while the vector $r = y - F\hat{x}$ can be efficiently updated from one iteration to the next, gMP with $T$ steps has roughly $T$ times the computational requirements of top-$k$.[1] Furthermore, we will find in practice that only $T = 3$ iterations are enough to significantly improve over top-$k$, even as the size of the inference problem varies over orders of magnitude.

The idea of gMP is closely related to well-known techniques in compressive sensing. For instance, the matching pursuit (Mallat & Zhang, 1993) and orthogonal matching pursuit (Tropp & Gilbert, 2007) algorithms infer a sparse signal $x$ by greedily selecting, one at a time, the column of $F$ most correlated with the current residual. (Matching pursuit subtracts the chosen codeword's contribution from the residual, while orthogonal

---

[1]In practice, we foresee that other optimizations are possible; for instance, a subset of the columns of $F$ whose correlations with $y$ are below a certain threshold could be excluded, which would eliminate the need to compute all of the coefficients of the product $F^T r$ in subsequent iterations.

matching pursuit re-fits the residual against all selected codewords at each step.) Meanwhile, the technique of selecting multiple codewords at once was proposed in Wang et al. (2012) as an extension to orthogonal matching pursuit. The resulting algorithm was called generalized orthogonal matching pursuit (gOMP). gMP can be understood as a variant of gOMP specialized to the case where the coordinates of $x$ are known to belong to $\{0, 1\}$.

## 3 Information Theory Bounds

To understand the minimum dimension $d$ required for a superposition code with parameters $(N, k)$ to be decodable by a given method, it will be helpful to use the point of view of information theory.

Let us first consider the case $k = 1$, meaning that the latent variable $X$ to be encoded is just a symbol drawn from an alphabet of size $N$. What is the minimum dimension $d$ for which each value of $X$ can be represented by a vector $Y \in \mathbb{R}^d$? When the vector $Y$ is stored exactly, the answer to our question simply depends on the number of values each coefficient can take. If the coefficients of $Y$ are 16-bit floating-point numbers, then each coefficient can store (nearly) $2^{16}$ distinct numbers, and overall we need only about $d_{\min} \approx \frac{1}{16} \log_2 N$ dimensions. Applying the same reasoning to the case where $X$ is a $k$-sparse subset suggests we may need only $\frac{1}{16} \log_2 \binom{N}{k}$ dimensions. However, this is a very optimistic prediction; in practice the ratio $d / \log_2 \binom{N}{k}$ will need to be significantly larger than $1/16$.

In the presence of exogenous noise, information theory provides a simple explanation for this kind of limitation. Specifically, suppose $Z \in \mathbb{R}^d$ is a vector of independent Gaussians each of variance $P$, and consider the problem of recovering $X$ from $Y + Z$. The following is then a very well-known (but remarkable) result of information theory.

**Proposition 3** (A Shannon limit). *Let the random variable $X \in [N]$ be uniformly distributed and let $Z \in \mathbb{R}^d$ be independent Gaussian noise of variance $P$. Suppose there exists a pair of maps $F \colon [N] \to \mathbb{R}^d$ and $G \colon \mathbb{R}^d \to [N]$ so that*

$$G(F(X) + Z) = X$$

*with probability at least $(1 - p)$, and suppose the variance of each coordinate of $F(X)$ is bounded by 1. Then*

$$d \geq 2 \frac{(1 - p) \ln N - \ln 2}{\ln(1 + P^{-1})}.$$

Asymptotically for large $N$, this means that

$$\frac{d}{H} \geq (1 + o(1)) \frac{2}{\ln(1 + P^{-1})}$$

if there is any pair $(F, G)$ that transmits $X$ with probability $1 - o(1)$, where $H = \ln N$ is the entropy of $X$. In other words, to transmit information reliably in the presence of noise of power $P$, we need at least $2 / \ln(1 + P^{-1})$ dimensions per nat.

Conversely, it can be shown that this bound on the ratio $d/H$ is asymptotically tight. More specifically, for any $\epsilon > 0$ and for sufficiently large $N$, it suffices that

$$\frac{d}{H} \geq (1 + \epsilon) \frac{2}{\ln(1 + P^{-1})}$$

for there to exist some pair $(F, G)$ satisfying the conditions of Proposition 3 with $p$ arbitrarily close to 0. The same conclusions hold more generally under different models for noisy transmission, substituting $P^{-1}$ in general for a "signal-to-noise ratio" (SNR). For a reference on these topics, see e.g. Chapter 9 of Cover & Thomas (2006).

Although our toy example does not involve exogenous noise, it will be useful to understand the sample complexity $d$ of a superposition code in terms of the asymptotically attainable factor $d/H$, where $H = \ln \binom{N}{k}$ is the entropy of the set $X$ being encoded. Before proceeding to our main results, let us describe one intuitive

reason that the factor $d/H$ required for a superposition code to function may be comparable to the inverse capacity of a noisy channel.

Consider a vector $X \in \{0,1\}^N$ divided into two halves $(X^0, X^1)$, each of length $N/2$ and with $k/2$ uniformly random nonzero entries, and consider a dictionary $F \in \mathbb{R}^{d \times N}$ whose first and last $N/2$ columns, denoted $(F^0, F^1)$, are statistically independent. The superposition code for $X$ under the dictionary $F$ is

$$Y = FX = F^0 X^0 + F^1 X^1.$$

Now, suppose we estimate each half of the vector $X$ by a function $\hat{X}^i(Y, F^i)$ depending only on $Y$ and on the corresponding half $F^i$ of the dictionary. If the estimate $\hat{X}(Y, F) = (\hat{X}^0(Y, F^0), \hat{X}^1(Y, F^1))$ is reliable, then each estimate $\hat{X}^i$ is reliable under a certain noisy channel with signal-to-noise ratio of 1. By analogy with a Gaussian channel, we expect the inverse capacity of such a channel to be around $2/\ln(1+1) = 2/\ln 2$ dimensions per nat. Since the entropy of each vector $X^i$ is $\ln \binom{N/2}{k/2}$, we conclude roughly that

$$d \geq \frac{2}{\ln 2} \ln \binom{N/2}{k/2}.$$

Since the total entropy of $(X^0, X^1)$ is $H = 2 \ln \binom{N/2}{k/2}$, overall we have

$$\frac{d}{H} \geq \frac{1}{\ln 2}.$$

Roughly, this indicates that if a decoder for a superposition code can be separated into two pieces that do not share information, then it cannot read more than 1 **bit per dimension**. Intuitively, such isolation properties should hold in an approximate sense for many simple decoding methods. This informal prediction is corroborated by our main results; for instance, in Figure 4, decoding is unreliable for all methods when $d/\tilde{H}_2 < 1$ and $\eta > 0$.

## 4 Main Results

We are now ready to state our main results. We begin in Section 4.1 with our theoretical guarantees for MAP decoding and top-$k$. Our main conclusion is Corollary 2, which gives an asymptotic description of sample complexity in the sublinear regime $k \leq N^\eta$ in terms of the ratio $d/H$. In Section 4.2, we use the same framework to empirically compare the performance of MAP and top-$k$ decoding with gMP. Finally, in Section 4.3, we describe how our findings compare to known results in compressive sensing.

### 4.1 Theoretical Guarantees for One-Step Methods

For a given family of dictionaries $F_{N,d} \in \mathbb{R}^{d \times N}$, consider the problem of recovering a random $k$-element subset $X$ from a $d$-dimensional superposition code. We will write

$$b(d, k, N; \tau), \quad b(d, k, N; \text{top-}k)$$

for the respective failure probabilities of threshold decoding at level $\tau$ and top-$k$ decoding.[2] In our theoretical results, we exclusively consider Rademacher dictionaries. Naturally we have $b(d, k, N; \text{top-}k) \leq b(d, k, N; \tau)$, since if a given code $Y$ is accurately decoded by threshold decoding at any level $\tau$, it is also decoded by top-$k$.

Our first result provides an asymptotic constraint on the dimension $d$ required for either of these values to converge to 0 for large $N$.

**Proposition 4.** *Let $C > 0$ be arbitrary. Over any regime where*

$$d \leq Ck \ln N, \quad \omega(1) \leq k < N/2,$$

*for Rademacher dictionaries it holds that*

$$\liminf_{N \to \infty} b(d, k, N; \text{top-}k) \geq 1 - \frac{C}{2}.$$

---

[2]The letter $b$ is a mnemonic for "bad event."

Here, $\omega(1)$ denotes any function that diverges to infinity for large $N$. A particular consequence of this result is that, in any regime where $k = \omega(1)$, $k < N/2$, and $b(d,k,N;\text{top-}k)$ converges to 0, for any $\epsilon > 0$ we must have $d \geq (2-\epsilon)k\ln N$ for sufficiently large $N$. Informally, we can say that top-$k$ (and hence MAP decoding) can only be guaranteed to succeed with high probability when $d \geq 2k\ln N$. Our proof (available in Appendix E) is based on an information-theoretic argument, similar in character to Proposition 3.

Now, in a regime where $k \geq \epsilon N$ for some $\epsilon > 0$, the entropy of a $k$-element subset of $[N]$ is

$$H = \ln \binom{N}{k} \leq k \ln \left( \frac{eN}{k} \right) = O(N).$$

This is asymptotically larger than the sample complexity of $2k\ln N$ implied by Proposition 4. In this regime, often called a regime of *linear sparsity*, we can informally say that the amount of information that top-$k$ can read from each dimension of the code converges to 0.

It is natural to ask whether top-$k$ can be "information-efficient" in some sparser regime. By our discussion, we would need $H/(k\ln N)$ to be bounded strictly above 0. Since

$$\frac{H}{k\ln N} = \frac{k\ln N - k\ln k + O(k)}{k\ln N} = 1 - \frac{\ln k}{\ln N} + O\left( \frac{1}{\ln N} \right),$$

we conclude that top-$k$ can only be information-efficient when $\ln k/\ln N$ is bounded strictly below 1. This motivates the introduction of a parameter $\eta = \ln k/\ln N$, which we call the sparsity exponent. In compressive sensing, when $k \leq N^\eta$ for some fixed $\eta < 1$, we say we are in a regime of *sublinear sparsity*. Our next result gives a sufficient condition for threshold decoding to succeed under sublinear sparsity.

**Proposition 5.** *For $\eta \in (0,1]$, let $k \leq N^\eta$. Then for any $\epsilon > 0$, over the regime*

$$d \geq (1+\epsilon)2(1+\sqrt{\eta})^2 k\ln N, \quad \tau \in \left[ \frac{1}{2} + (1-\eta)\frac{k\ln N}{d}, (1+\sqrt{\eta})^{-1} \right],$$

*for Rademacher dictionaries it holds that*

$$\lim_{N\to\infty} b(d,k,N;\tau) = 0.$$

In the extreme case where $\eta = 1$, the condition $k \leq N^\eta$ is vacuous and overall Proposition 5 lets us conclude that $d \geq (1+\epsilon)8k\ln N$ dimensions are enough for threshold decoding to succeed with threshold $\tau = 1/2$ for any $k$ as $N \to \infty$. For smaller values of $\eta$, Proposition 5 guarantees a lower sample complexity when the threshold $\tau$ is strictly larger than $1/2$. Note that the lower bound on $\tau$ in this statement is identical to the threshold $\tau_{\text{MAP}}$ motivated in Section 2.1, after substituting $\eta = \ln k/\ln N$. See Appendix F for a proof of Proposition 5.

Figure 3 shows how the critical value $d_{\text{crit}} = 2(1+\sqrt{\eta})^2 k\ln N$ appearing in the statement of Proposition 5 compares with empirical performance of MAP thresholding and top-$k$ for finite values of $N$ and $k$. Note that the statement of our result is asymptotic, and so does not give guarantees on the performance of either decoding method for finite $(k,N)$. (More explicit statements can be derived from Lemma 3 of Appendix F, which provides an explicit upper bound on $b(d,k,N;\tau)$.) However, $d_{\text{crit}}$ serves empirically as a good rule of thumb; for $k < 64$ and for values of $N$ ranging over several orders of magnitude, $d_{\text{crit}}$ can be used to reliably predict the performance of both MAP thresholding and top-$k$. In particular, although we find top-$k$ is noticeably more reliable, the sample complexities of both methods are similar and grow as a function of $k$ and $N$ in a similar way.

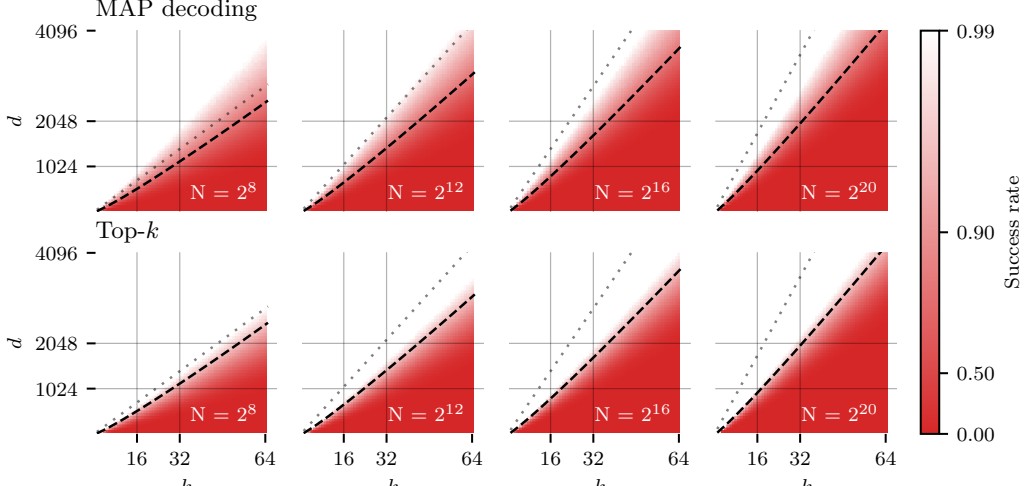

Figure 3: Empirical performance of MAP thresholding and top-$k$ decoding at recovering a $k$-element subset of $[N]$ from a $d$-dimensional superposition code using a Rademacher dictionary. The dashed line shows the expression $d = 2(1 + \sqrt{\eta})^2 k \ln N$ appearing in Proposition 5, evaluated with $\eta = \ln k / \ln N$, and the dotted line shows the $\eta$-independent upper bound of $d = 8k \ln N$.

Now, we return to the information-theoretic point of view. When $k \sim N^\eta$, the approximation $\ln \binom{N}{k} = k \ln(N/k) + O(k)$ shows that

$$\frac{H}{k \ln N} = \frac{\ln \binom{N}{k}}{k \ln N} = 1 - \eta + O\left(\frac{1}{\ln N}\right).$$

In this regime, the critical values of $2k \ln N$ and $2(1+\sqrt{\eta})^2 k \ln N$ in the respective statements of Proposition 4 and Proposition 5 satisfy

$$\lim_{N \to \infty} \frac{2k \ln N}{H} = \frac{2}{1 - \eta}, \quad \lim_{N \to \infty} \frac{2(1 + \sqrt{\eta})^2 k \ln N}{H} = \frac{2(1 + \sqrt{\eta})^2}{1 - \eta} = \frac{2(1 + \sqrt{\eta})}{1 - \sqrt{\eta}}.$$

Thus, we can summarize these two results in the following way.

**Corollary 2.** *Let $\{top\text{-}k, MAP\}$ denote the methods of top-k decoding and threshold decoding at level $\tau_{MAP}$. Where $m$ denotes a method and $\eta \in (0, 1)$, let $C(\eta; m)$ be the infimum of all constants $C$ such that, over all regimes satisfying $\omega(1) \le k \le N^\eta$ and $d \ge CH$,*

$$\lim_{N \to \infty} b(d, k, N; m) = 0.$$

*Then, for Rademacher dictionaries,*

$$\frac{2}{1 - \eta} \le C(\eta; top\text{-}k) \le C(\eta; MAP) \le \frac{2(1 + \sqrt{\eta})}{1 - \sqrt{\eta}}.$$

The first and last inequalities above follow as direct applications of Proposition 4 and Proposition 5, and the inequality $C(\eta; \text{top-}k) \le C(\eta; \text{MAP})$ follows from our remark that $b(d, k, N; \text{MAP}) \ge b(d, k, N; \text{top-}k)$.

Informally, we may interpret $C(\eta; m)$ as the number of dimensions required to transmit each nat of information as $N \to \infty$ and $k \sim N^\eta$. In the limit $\eta \to 0$, note that these two expressions match; we find that 2 dimensions per nat (around 1.4 dimensions per bit) is necessary for top-$k$ decoding and sufficient for MAP decoding. Thus, these methods have very similar sample complexities when the vector $X$ is sufficiently sparse. As $\eta \to 1$, the sample complexities of both methods diverge to $+\infty$.

### 4.2 Numerical Experiments

Now, let us compare our theoretical results with empirical performance of top-$k$, MAP decoding, and gMP.

In view of Corollary 2, we will be interested in studying the values of $d/H$ at which a given decoding method succeeds for a given value of $\eta = \ln k / \ln N$. To estimate the denominator $H = \ln \binom{N}{k}$ we will use the approximation $\tilde{H} = k \ln(eN/k)$. When $k \leq N^\eta$, it holds that

$$\tilde{H} = k \ln(eN/k) = \ln \binom{N}{k} + O(N^{2\eta-1}),$$

and the remainder $O(N^{2\eta-1})$ vanishes for $\eta < 1/2$. (See Appendix C for a derivation.) For example, when $N = 2^{16}$ and $k \leq 5000$, numerical evaluation shows that the relative error of this approximation is smaller than 1.4%. We will write $\tilde{H}_2 = \tilde{H}/\ln 2$ for the expression of $\tilde{H}$ in bits.

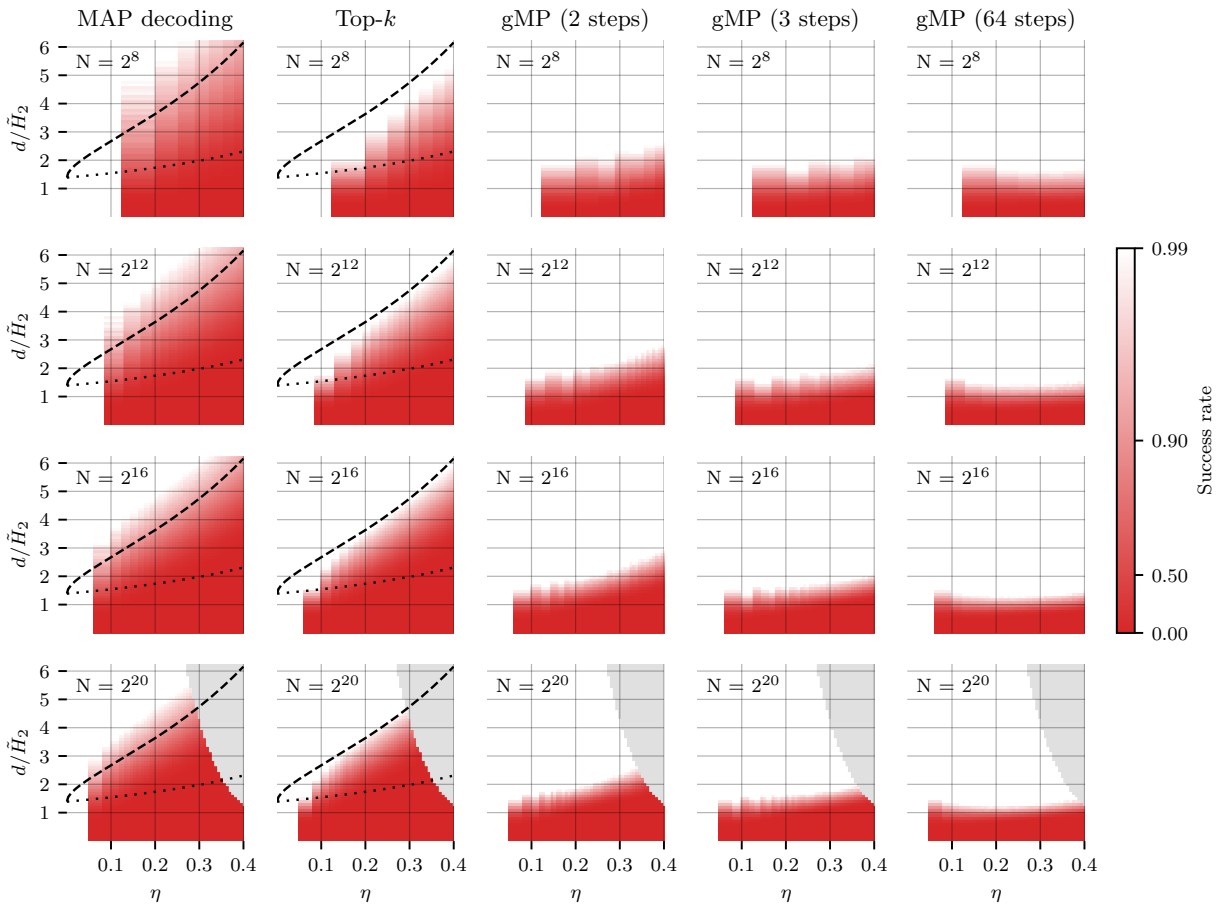

Figure 4: **How many dimensions do we need to store one bit of information?** We graph empirical performance of MAP thresholding, top-$k$, and gMP at exactly decoding a superposition code as a function of approximate inverse bitrate $d/\tilde{H}_2 = d/(k \log_2(eN/k))$ and sparsity exponent $\eta = \ln k / \ln N$. The dashed and dotted lines, respectively, plot the upper and lower bounds appearing in Corollary 2.

Figure 4 shows the empirical performance of different decoding methods in terms of the ratio $d/\tilde{H}_2$ and the exponent $\eta = \ln k / \ln N$. We have taken $N$ to range over $\{2^8, 2^{12}, 2^{16}, 2^{20}\}$ and $\eta \in (0, 0.4]$. For each value of $(N, \eta)$, we choose $k$ as the closest integer to $N^\eta$ and compute the probability of successful recovery for different values of $d$ by empirical simulation over a large batch of problem instances. No results are reported where $\eta$ is small enough to make $k$ round to 1, since in this case the inference problem is trivial. Results on the bottom row are truncated where they would require a dictionary with dimensions larger than $4096 \times 2^{20}$.

The two left-most columns of Figure 4 show empirical performance of MAP decoding and top-$k$, previously reported in Figure 3, now graphed in the $(\eta, d/\tilde{H}_2)$ coordinate system. The upper and lower bounds on the asymptotic constants $C(\eta; m)$ indicated in Corollary 2 are represented by dashed and dotted lines respectively. We find that empirical sample complexity of both methods can be roughly predicted by the upper bound, in agreement with Figure 3.

The next three columns of Figure 4 show the performance of generalized matching pursuit (gMP). We find that, even with $T = 3$ steps, gMP succeeds reliably with $d/\tilde{H}_2 > 2$ over the range $\eta \in (0, 0.35)$. Increasing the step parameter further shows markedly diminishing returns.

With the values $N = 2^{20}$ and $k = 64$ we referenced from Gao et al. (2024), we have $\eta = \ln k / \ln N = 0.3$. At this value of $\eta$, we find that slightly over 4 dimensions per bit are required to reliably decode a superposition code using the top-$k$ decoder, even as $N$ ranges from $2^8$ to $2^{20}$. Meanwhile, gMP with $T = 3$ steps requires only 2 dimensions per bit.

Although our results thus far have applied only on Rademacher dictionaries, we find that our experimental results are essentially unmodified when using spherical dictionaries (Appendix G). Code for our experiments and graphs is available at `https://github.com/cgadski/bitrate_paper`.

### 4.3 Comparison with Compressive Sensing

We now briefly compare our findings with known results from the compressive sensing literature. Note that compressive sensing typically concerns signals whose non-zero coordinates take arbitrary real values, often observed with measurement noise, whereas our toy model asks for exact recovery of a binary vector from noiseless measurements.

First, consider the regime of linear sparsity, where $k \sim sN$ for a given $s \in (0, 1)$. A fundamental result of compressive sensing is that, as $N \to \infty$, $k$-sparse vectors $x \in \mathbb{R}^N$ can be exactly recovered from certain random projections $y = Fx$ with high probability by solving the linear program

$$\text{minimize } \|x\|_1 \quad \text{subject to } Fx = y,$$

so long as the "aspect ratio" $d/N$ of the matrix $F$ exceeds a certain threshold $\delta(s) < 1$; for instance, see Donoho & Tanner (2010). In contrast, Proposition 4 shows that top-$k$ requires $d = \Omega(N \ln N)$ dimensions in our setting. Unsurprisingly, we find that one-step methods have comparatively poor sample complexity in this regime.

Under sublinear sparsity $k \sim N^\eta$, the sample complexity described in compressive sensing is typically on the order of $k \ln(N/k)$. For instance, Reeves et al. (2019) proved that, in a model essentially identical to ours except for additive Gaussian noise of variance $\sigma^2$, maximum likelihood decoding exhibits an "all-or-nothing" transition at

$$d^* = \frac{2k \ln(N/k)}{\ln(1 + k/\sigma^2)}.$$

Iterative methods approaching such information-theoretic limits under sublinear sparsity have been analyzed by Truong (2023); Takeuchi (2024). Actually, since $k \ln(N/k)$ is asymptotic to the entropy

$$H = \ln \binom{N}{k} = k \ln(N/k) + O(k) = (1 + o(1)) \, k \ln(N/k)$$

when $N/k \to \infty$, our results show that one-step methods attain sample complexities of the same order, with a constant factor depending on $\eta$.

The closest prior analysis of one-step methods we are aware of is Bajwa et al. (2010), which shows (in a noisy setting, under certain general conditions on the dictionary) that one-step thresholding recovers the support of a sparse signal when $d \geq C \, k \ln N$, where $C$ is a constant independent of the sparsity regime. However, our results strongly suggest that no bound of this form can be sharp. Indeed, as $\eta \to 0$ we have found that the necessary bound $d \geq (1 + o(1))2k \ln N$ of Proposition 4 agrees with the sufficient condition of Proposition 5, but empirically Figure 4 shows that for moderate values of $\eta$ a larger constant factor is required.

## 5  Are Random Dictionaries Optimal?

In the toy scenario studied in this work, the dictionary $F$ is randomly generated. Of course, the dictionaries employed by neural networks may have special structure. It is natural to ask whether our findings would be significantly different if the dictionary were specially designed in some way.

One natural measurement for the coherence of a dictionary is the mean squared interference of codewords

$$\xi(F) = \binom{N}{2}^{-1} \sum_{i<j} \langle f_i, f_j \rangle^2.$$

Indeed, if we assume an isotropic distribution of codewords, from the point of view of Section 2 the mean squared interference controls the average signal-to-noise ratio of the "filters" $\lambda_i(Y) = \langle f_i, Y \rangle$.

An important result in coding theory known as the Welch bound (Welch, 1974) states that, for any dictionary $F \in \mathbb{R}^{d \times N}$ of unit norm codewords, we have

$$\sum_{i,j} \langle f_i, f_j \rangle^2 \geq \frac{N^2}{d}.$$

This can be derived by viewing the sum of squared inner products above as the sum of squared eigenvalues $\lambda_i^2$ of the Gram matrix $F^T F$ and applying a Cauchy–Schwarz inequality, using the fact that $\sum_i \lambda_i = N$.

In terms of mean squared interference $\xi(F)$, the Welch bound means that

$$\xi(F) \geq \frac{1}{N^2 - N} \left( \frac{N^2}{d} - N \right) = \frac{1}{d} \left( 1 - \frac{d}{N} \right) \left( 1 + \frac{1}{N-1} \right).$$

Therefore, in a limit where $N \to \infty$ and $d/N \to 0$, we have $\xi(F) \geq (1 + o(1))/d$. However, it is easy to check that $\mathrm{E}\,\xi(F) = 1/d$ for both Rademacher and spherical dictionaries, and moreover $\xi(F)$ concentrates tightly around this value as $N$ and $d$ grow. So, in such a limit, the minimum value of $\xi(F)$ does not improve meaningfully over the value attained by a random dictionary. It stands as a conjecture that, in the sublinear regime, the performance of MAP thresholding and top-$k$ is not significantly improved by any family of structured dictionaries.

## 6  Summary and Conclusions

Over the course of computation, computer programs use memory to record intermediate results. Similarly, residual vectors within a deep neural network must encode intermediate "features" relevant to the task being carried out. Reverse engineering representations used by large neural networks is a broad goal considered within mechanistic interpretability. (See, for example, Section 2.1 of Sharkey et al. (2025).)

A common assumption, familiar from the point of view of connectionism, is that these vectors should be modeled as distributed representations. Roughly, this means that different underlying features $x_i$ are simultaneously represented by patterns of activity taking place over a whole vector $y$. Unfortunately, inferring such a representation can be a difficult statistical problem. Sparse autoencoders (SAEs) are one recently proposed solution.

Given a model expressing $y$ in terms of a vector $x$ of latent features, consider the subproblem of inferring $x$ from $y$. One possible approach is to estimate each feature $x_i$ independently, approximating the contribution of the remaining features as an idealized noise. The encoder layers of existing sparse autoencoders can be understood as estimates of this form. We call these "one-step estimates." Of course, it may be possible to infer $x$ more reliably through an iterative approach, where the running estimate for one factor $x_j$ is used to improve the estimates of other factors $x_i$. What is not obvious a priori is *how much better* we can expect an iterative approach to be. Are iterative estimates necessary to decode distributed representations?

This work studies basic mathematical phenomena that may help answer this question. In a specific toy scenario modelling extremely sparse vectors, we showed that the sample complexity of "one-step methods" is

a constant factor of the Shannon limit. More specifically, the number of dimensions required to transmit each bit of information can be predicted as a function of a certain parameter $\eta = \ln k / \ln N$ measuring sparsity of the latent vector $x$. On the other hand, we showed experimentally that it is possible to significantly reduce sample complexity by paying only a moderate constant-factor computational expense.

The most obvious result of our findings is to suggest that the performance of sparse autoencoders may be meaningfully improved by using slightly more computation to solve the sparse reconstruction problem.[3] However, we hope that the information-theoretic vantage point of our work is broadly useful in future work that attempts to model distributed representations.

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

## A  Matched Filters

Consider the problem of inferring a scalar $S$ from the sum

$$Y = Sf + Z$$

where $f \in \mathbb{R}^d$ is fixed and $Z \in \mathbb{R}^d$ is a Gaussian variable independent from $S$. Suppose for simplicity that $Z$ has non-singular covariance matrix $\Sigma \in \mathbb{R}^{d \times d}$, so that $\ln p(z) = C_\Sigma - \|z\|_\Sigma^2/2$ where $C_\Sigma$ depends only on $\Sigma$ and $\|z\|_\Sigma^2 = z^T \Sigma^{-1} z$. In this appendix, we briefly recall how the posterior on $S$ can be written in terms of a certain linear function $\lambda(Y)$ of $Y$, which we call a matched filter.

Using that $S$ is independent from $Z$, observe first that

$$
\begin{aligned}
\ln p(Y = y \mid S = s) &= \ln p(Z = y - sf \mid S = s) \\
&= \ln p(Z = y - sf) \\
&= C_\Sigma - \frac{1}{2}\|y - sf\|_\Sigma^2.
\end{aligned}
\tag{5}
$$

Developing the norm $\|y - sf\|_\Sigma^2$ as a function of $s$ and completing a square gives

$$
\begin{aligned}
\|y - sf\|_\Sigma^2 &= \|y\|_\Sigma^2 - 2\langle y, f\rangle_\Sigma s + \|f\|_\Sigma^2 s^2 \\
&= \|y\|_\Sigma^2 - \frac{\langle y, f\rangle_\Sigma^2}{\|f\|_\Sigma^2} + \|f\|_\Sigma^2 \left(s - \frac{\langle y, f\rangle_\Sigma}{\|f\|_\Sigma^2}\right)^2,
\end{aligned}
$$

where $\langle v, w\rangle_\Sigma = v^T \Sigma^{-1} w$ is the inner product associated with $\|-\|_\Sigma$. Overall,

$$\ln p(Y = y \mid S = s) = C(y, \Sigma) - \frac{\|f\|_\Sigma^2}{2} \left(s - \frac{\langle y, f\rangle_\Sigma}{\|f\|_\Sigma^2}\right)^2$$

for a certain $C(y, \Sigma)$ not depending on $s$. The maximum likelihood estimate for $S$ conditional on $Y$ is therefore

$$\hat{S} = \frac{\langle f, Y\rangle_\Sigma}{\|f\|_\Sigma^2}.$$

We call this linear function of $Y$ the matched filter for $S$, and denote it by $\lambda$.

We can also write the posterior on $S$ in terms of the matched filter, since

$$
\begin{aligned}
\ln p(S = s \mid Y = y) &= \ln p(Y = y \mid S = s) - \ln p(Y = y) + \ln p(S = s) \\
&= C(y, \Sigma) - \ln p(Y = y) + \ln p(S = s) - \frac{\|f\|_\Sigma^2}{2}\left(s - \lambda(y)\right)^2.
\end{aligned}
\tag{6}
$$

Finally, note that the norm $\|f\|_\Sigma^2$ is exactly the inverse variance of the noise $Z$ under the matched filter. Indeed, since for any $w \in \mathbb{R}^d$ we have

$$\text{Var}\,\langle w, Z\rangle = w^T \Sigma w,$$

applying this formula with $w = \Sigma^{-1} f$ gives

$$
\begin{aligned}
\text{Var}\,\lambda(Z) = \text{Var}\,\frac{\langle f, Z\rangle_\Sigma}{\|f\|_\Sigma^2} &= \frac{1}{\|f\|_\Sigma^4}\,\text{Var}\,\langle \Sigma^{-1}f, Z\rangle \\
&= \frac{f^T \Sigma^{-1} \Sigma \Sigma^{-1} f}{\|f\|_\Sigma^4} = \frac{1}{\|f\|_\Sigma^2}.
\end{aligned}
$$

In the parlance of signal processing, we would call $\|f\|_\Sigma^2$ the "signal-to-noise ratio" $\rho$ of the filter $\lambda$, since it measures the ratio between the magnitude $\lambda(f) = 1$ of its sensitivity to the signal $S$ and the magnitude $\text{Var}\,\lambda(Z)$ of its sensitivity to the noise.

# B Derivation of the Maximum a Posteriori Threshold

For a given index $i$, recall that $Z_i = \sum_{j \neq i} X_j f_j$ is the nuisance term in the problem of estimating $X_i$ from the superposition code

$$Y = FX = X_i f_i + Z_i.$$

In Section 2.1, we stated that this estimation can be performed by replacing $Z_i$ with a Gaussian variable $\tilde{Z}$. We claimed that the maximum a posteriori estimate for $X_i$ under the simplified model then takes the form

$$\hat{X}_i(Y) = \begin{cases} 1 \colon \lambda_i(Y) \geq \tau \\ 0 \colon \text{otherwise}, \end{cases} \tag{7}$$

for a certain threshold $\tau \in \mathbb{R}$. In this section, we detail this derivation.

First, let us determine the appropriate variance for $\tilde{Z}$ by calculating the variance of $Z_i$. Suppose that the codewords $F_i$ are centered and isotropically distributed random variables, meaning $\mathrm{E}\, F_i = 0$ and $\mathrm{Var}\, F_i = \alpha I$ for some scalar $\alpha > 0$. This is true for Rademacher and spherical dictionaries, for instance. If in addition the codewords are unit vectors, then necessarily we have $\alpha = 1/d$ since

$$1 = \mathrm{E}\, \|F_i\|^2 = \mathrm{tr}\, \mathrm{Var}\, F_i = d\alpha.$$

Conditional on $X_i = 0$, $Z_i$ is a sum of $k$ independent codewords, and so its coordinates have variance $k/d$. Meanwhile, conditional on $X_i = 1$, the coordinates of $Z_i$ have variance $(k-1)/d$. Since $\mathrm{P}(X_i = 1) = k/N$, overall $\mathrm{Var}\, Z_i = \beta I$ with

$$\beta = \frac{N-k}{N} \frac{k}{d} + \frac{k}{N} \frac{k-1}{d} = \frac{k}{Nd} \big[(N-k) + (k-1)\big] = \frac{N-1}{N} \frac{k}{d}.$$

Thus, to best approximate $Z_i$, it is natural to define $\tilde{Z}$ to be an isotropic centered Gaussian where each coordinate has variance $(N-1)k/Nd$.

Because $\tilde{Z}$ is isotropic and $\|f_i\| = 1$, it follows from Appendix A that the maximum likelihood estimator for $X_i$ is simply the inner product $\lambda_i(Y) = \langle f_i, Y \rangle$. The posterior on $X_i$ under our simplified Gaussian model is

$$\ln \mathrm{P}(X_i = x \mid Y = y) = C + \ln \mathrm{P}(X_i = x) - \frac{\rho}{2}\left(x - \lambda_i(y)\right)^2,$$

where $C$ does not depend on $x$ and $\rho$ is the signal-to-noise ratio

$$\rho = \left(\mathrm{Var}\, \langle f_i, \tilde{Z} \rangle\right)^{-1} = \left(\frac{(N-1)k}{Nd} \|f_i\|^2\right)^{-1} = \frac{N}{N-1} \frac{d}{k}.$$

The log odds that $X_i = 1$ is therefore

$$\ln \frac{\mathrm{P}(X_i = 1 \mid Y = y)}{\mathrm{P}(X_i = 0 \mid Y = y)} = \ln \frac{\mathrm{P}(X_i = 1)}{\mathrm{P}(X_i = 0)} - \frac{\rho}{2}\left[(1 - \lambda_i(y))^2 - \lambda_i(y)^2\right] = \ln \frac{\mathrm{P}(X_i = 1)}{\mathrm{P}(X_i = 0)} + \rho\left(\lambda_i(y) - \frac{1}{2}\right).$$

The maximum a posteriori estimate for $X_i$ equals 1 exactly when this quantity is non-negative, i.e. when $\langle f_i, Y \rangle$ exceeds the threshold

$$\tau = \frac{1}{2} - \frac{1}{\rho} \ln \frac{\mathrm{P}(X_i = 1)}{\mathrm{P}(X_i = 0)}.$$

Substituting the prior that $\mathrm{P}(X_i = 1) = k/N$ gives

$$\tau = \frac{1}{2} - \rho^{-1} \ln \frac{k}{N-k} = \frac{1}{2} + \frac{N-1}{N} \frac{k}{d} \ln \frac{N-k}{k}. \tag{8}$$

Using that

$$\ln \frac{N-k}{k} = \ln \frac{N}{k} + \ln\left(1 - \frac{k}{N}\right) = \ln N - \ln k + O(k/N),$$

we can expand (8) as

$$
\begin{aligned}
\tau &= \frac{1}{2} + \frac{N-1}{N}\frac{k}{d}\ln\frac{N-k}{k} \\
&= \frac{1}{2} + \left(1 + O\left(\tfrac{1}{N}\right)\right)\frac{k}{d}\left(\ln N - \ln k + O\left(\tfrac{k}{N}\right)\right) \\
&= \frac{1}{2} + \frac{k}{d}\left(\ln N - \ln k\right) + O\left(\frac{k\ln N}{Nd}\right) + O\left(\frac{k^2}{Nd}\right).
\end{aligned}
$$

Since we study a regime where $d = \Omega(k\ln N)$ and $k \le N^\eta$ for $\eta < 1$, both terms are $O(k/N)$ and in particular are very small in practice. Overall, we are left with the simplified thresholding level

$$
\tau_{\mathrm{MAP}} = \frac{1}{2} + \frac{k}{d}\left(\ln N - \ln k\right) = \frac{1}{2} + \left(1 - \frac{\ln k}{\ln N}\right)\frac{k\ln N}{d}.
$$

## C   Estimates for the Binomial Coefficient

To estimate $\ln\binom{N}{k}$, it is helpful to first remember the elementary inequalities

$$
\left(\frac{N}{k}\right)^k \le \binom{N}{k} \le \left(\frac{eN}{k}\right)^k.
$$

(For a reference, see Appendix C of Cormen (2009).) It follows that $k\ln(N/k) \le \ln\binom{N}{k} \le k\ln(eN/k)$, so

$$
\ln\binom{N}{k} = k\ln(N/k) + O(k).
$$

When $k \ll N$, the upper bound $\ln\binom{N}{k} \le k\ln(eN/k)$ is a better approximation; in fact

$$
\ln\binom{N}{k} = k\ln(eN/k) + O(k^2/N).
$$

In terms of $k = N^\eta$ the remainder is $O(N^{2\eta-1})$, which vanishes for $\eta < 1/2$.

One heuristic justification for this finer estimate is to consider a random subset $Y \subseteq [N]$ where each element is included independently with probability $s = k/N$. Then, where

$$
h(s) = -s\ln s - (1-s)\ln(1-s) = -s\ln s + s + O(s^2)
$$

is the binary entropy function, the entropy of $Y$ is

$$
\begin{aligned}
H(Y) &= h(s)N = -sN\ln s + sN + O(s^2 N) \\
&= k\ln(eN/k) + O(k^2/N).
\end{aligned}
$$

We expect that for large $N$ the number of elements in $Y$ concentrates sharply around $k$, and so $H(Y)$ should be close to $\ln\binom{N}{k}$.

Indeed, this can be proven using the Stirling approximation

$$
\ln n! = n\ln n - n + (1/2)\ln(2\pi n) + O(n^{-1}).
$$

Substituting into the binomial gives

$$
\begin{aligned}
\ln\binom{N}{k} &= \ln N! - \ln k! - \ln(N-k)! \\
&= k\ln(N/k) + (N-k)\ln(1 + k/(N-k)) + O(1/N),
\end{aligned}
$$

and finally the Taylor approximation $\ln(1 + k/(N - k)) = k/N + O(k^2/N^2)$ yields

$$\ln \binom{N}{k} = k \ln(N/k) + (N - k)[k/N + O(k^2/N^2)] + O(1/N)$$

$$= k \ln(N/k) + k(N - k)/N + O(k^2/N)$$
$$= k \ln(N/k) + k(1 - k/N) + O(k^2/N)$$
$$= k \ln(N/k) + k + O(k^2/N)$$
$$= k \ln(eN/k) + O(k^2/N).$$

## D    Basic Results on Random Vectors

The following is a well-known estimate for the tail of a Gaussian.

**Proposition 6.** *When $Z$ is a unit Gaussian, we have*

$$\ln \mathrm{P}(Z \geq a) = -\frac{1}{2}a^2 - \ln a - \frac{1}{2} \ln 2\pi + o(1)$$

*as $a \to +\infty$.*

In fact, the leading-order term $-\frac{1}{2}a^2$ is an upper bound on $\ln \mathrm{P}(Z \geq a)$. One simple way to establish such estimates on tail probabilities is via the Chernoff inequality: for any random variable $X$ and $\lambda > 0$,

$$\mathrm{P}(X \geq a) \leq e^{-\lambda a}\, \mathrm{E}\, e^{\lambda X} = \exp(-\lambda a + K_X(\lambda))$$

where $K_X(\lambda) = \ln \mathrm{E}\, e^{\lambda X}$ is the cumulant generating function. In general, putting $\lambda = a$ in the Chernoff inequality for a random variable $X$ with $K_X(\lambda) \leq \lambda^2/2$ gives

$$\mathrm{P}(X \geq a) \leq \exp(-a^2/2),$$

which is called a sub-Gaussian tail bound.

Let $X_1, \ldots, X_k$ be independent Rademacher random variables, each uniformly distributed on $\{-1, 1\}$. Then $K_{X_i}(\lambda) = \ln \cosh \lambda \leq \lambda^2/2$, so

$$K_{R_k}(\lambda) = \sum_{i=1}^{k} K_{X_i}(\lambda) \leq k\lambda^2/2$$

where $R_k = X_1 + \cdots + X_k$. This proves the following.

**Proposition 7** (Chernoff bound for Rademacher sums)**.** *For $R_k$ a sum of $k$ independent Rademacher variables and $t > 0$,*

$$\mathrm{P}(R_k \geq t) \leq \exp\left(-\frac{t^2}{2k}\right).$$

This is enough to prove Proposition 1, which we restate here for convenience.

**Proposition.** *For $\mu > 0$ and $p > 0$, let $d \geq 2\mu^{-2}(2 \ln N + \ln p^{-1})$, and let*

$$\{F_1, \ldots, F_N\} \subseteq \{-1/\sqrt{d}, 1/\sqrt{d}\}^d$$

*be random vectors with independent, uniformly distributed entries. Then $|\langle F_i, F_j \rangle| < \mu$ for all $i \neq j$ with probability at least $(1 - p)$.*

*Proof.* Each inner product $\langle F_i, F_j \rangle$ is distributed like $R_d/d$ where $R_d$ is a sum of $d$ Rademacher variables. By the Chernoff bound, $\mathrm{P}(\langle F_i, F_j \rangle \geq \mu) = \mathrm{P}(R_d \geq d\mu) \leq \exp(-d\mu^2/2)$. By symmetry and a union bound over all $\binom{N}{2} < N^2/2$ pairs,

$$\mathrm{P}(\exists i \neq j : |\langle F_i, F_j \rangle| \geq \mu) \leq N^2 \exp(-d\mu^2/2).$$

This is at most $p$ when $d \geq 2\mu^{-2}(2 \ln N + \ln p^{-1})$. $\qquad\square$

The interested reader should also compare this result to the Johnson-Lindenstrauss lemma, which is proved in a very similar way. See Dasgupta & Gupta (2003) for a proof, or the last section of Foucart & Rauhut (2013) for a discussion of the JL lemma with some broader context.

# E   Proof of Proposition 4

In Section 3, we recalled that information theory can establish bounds on the amount of information that can be transmitted over a noisy channel. These bounds are called channel capacities. For instance, when a random vector $Y \in \mathbb{R}^d$ with coordinates $Y_i$ satisfying $\mathrm{Var}(Y_i) \leq 1$ is corrupted by addition of a Gaussian vector $Z$ with independent coordinates each of variance $P$, the resulting "transmission" can convey at most $d \ln(1 + P^{-1})/2$ nats of information to a receiver. More precisely, in terms of mutual information, this means that

$$I(Y; Y + Z) \leq \frac{d}{2} \ln(1 + P^{-1})$$

so long as $Y$ and $Z$ are independent. For large $P$, note that $\ln(1 + P^{-1}) = P^{-1} + O(P^{-2})$, so in this case the channel capacity is approximately $d/(2P)$. Finally, it can be shown that $I(Y; Y + Z) = \frac{d}{2} \ln(1 + P^{-1})$ exactly when the coordinates of $Y$ are independent unit Gaussians. For a reference on these facts, we again suggest Cover & Thomas (2006).

Mutual information bounds of this form are the basic tool underlying our proof of Proposition 4. Specifically, we will use the following bound on the capacity attainable by a vector in superposition with Rademacher codewords. (Recall that, by our earlier definition of a Rademacher dictionary, Rademacher codewords are distributed uniformly over $\{-1/\sqrt{d}, 1/\sqrt{d}\}^d$.)

**Proposition 8.** *Let $Z \in \mathbb{R}^d$ be a sum of $k$ independent Rademacher codewords. Then for any random vector $Y \in \mathbb{R}^d$ supported on $\{-1/\sqrt{d}, 1/\sqrt{d}\}^d$ and independent from $Z$,*

$$I(Y; Y + Z) \leq d\left(\frac{1}{2k} + O(k^{-2})\right).$$

*Proof.* Because $Y$ and $Z$ are independent, $H(Y + Z \mid Y) = H(Z)$. This lets us rewrite the mutual information as a difference of entropies:

$$I(Y; Y + Z) = H(Y + Z) - H(Y + Z \mid Y) = H(Y + Z) - H(Z).$$

Since the coordinates of $Z$ are independent, $H(Z) = \sum_i H(Z_i)$. Furthermore, by subadditivity of entropy,

$$H(Y + Z) \leq \sum_i H(Y_i + Z_i),$$

with equality when the coordinates of $Y_i$ are independent. We will prove that, for each $i$,

$$H(Y_i + Z_i) - H(Z_i) \leq \frac{1}{2k} + O(k^{-2}).$$

Recall that $Z_i$ is a sum of $k$ independent uniform signs $\pm 1/\sqrt{d}$, while $Y_i$ is supported on $\{-1/\sqrt{d}, 1/\sqrt{d}\}$ and independent of $Z_i$.

First, we claim that $H(Y_i + Z_i)$ is maximized when $Y_i$ is uniform on $\{-1/\sqrt{d}, 1/\sqrt{d}\}$. To see why, parameterize the distribution of $Y_i$ by $\lambda \in [0, 1]$ such that $\mathrm{P}(Y_i = -1/\sqrt{d}) = \lambda$, and define the function $f \colon [0, 1] \to \mathbb{R}$ by

$$f(\lambda) = H(Y_i + Z_i).$$

Because entropy is a concave function of probabilities and $\mathrm{P}(Y_i + Z_i = x)$ is an affine function of $\lambda$ for each $x$, $f$ is concave. Furthermore, because $Z_i$ is distributed like $-Z_i$,

$$f(\lambda) = H(Y_i + Z_i) = H(-Y_i - Z_i) = H(-Y_i + Z_i) = f(1 - \lambda).$$

From these two facts, $f$ must attain its maximum value at $\lambda = 1/2$.

To bound $H(Y_i + Z_i) - H(Z_i)$, it therefore suffices to consider the case that $Y_i$ is uniform. Using independence again, it is convenient to regard this difference as a mutual information:

$$H(Y_i + Z_i) - H(Z_i) = H(Y_i + Z_i) - H(Y_i + Z_i \mid Y_i) = I(Y_i; Y_i + Z_i).$$

To compute $I(Y_i; Y_i + Z_i)$, observe that $Y_i + Z_i$ is a sum of $k + 1$ independent uniform signs $\pm 1/\sqrt{d}$. Conditional on $Y_i + Z_i = w$, exactly $(\sqrt{d}w + k + 1)/2$ of these $k + 1$ signs are positive, and by symmetry, each summand is equally likely to be among the positive ones. The posterior on $Y_i$ is therefore

$$\mathrm{P}\left(Y_i = 1/\sqrt{d} \mid Y_i + Z_i\right) = \frac{\sqrt{d}(Y_i + Z_i) + k + 1}{2(k+1)} = \frac{1}{2} + \frac{\sqrt{d}(Y_i + Z_i)}{2(k+1)}.$$

Putting $S = \sqrt{d}(Y_i + Z_i)/(2(k+1))$ and writing $h$ for the binary entropy function, we find that

$$I(Y_i; Y_i + Z_i) = H(Y_i) - H(Y_i \mid Y_i + Z_i) = \ln 2 - \mathrm{E}\left[h\left(\frac{1}{2} + S\right)\right].$$

Taking a series expansion of $h$ at $1/2$ gives

$$I(Y_i; Y_i + Z_i) = \mathrm{E}\left[2S^2 + O(S^4)\right].$$

Note that $S$ is distributed like $R_{k+1}/(2(k+1))$, where $R_{k+1}$ is a sum of $(k+1)$ Rademacher variables. Recall that the moments of this sum satisfy

$$\mathrm{E}[R_{k+1}^2] = k + 1, \quad \mathrm{E}[R_{k+1}^4] = O(k^2).$$

We conclude that

$$I(Y_i; Y_i + Z_i) = 2\,\mathrm{E}[S^2] + O(\mathrm{E}[S^4]) = \frac{1}{2(k+1)} + O(k^{-2}) = \frac{1}{2k} + O(k^{-2}).$$

$\square$

It will also be useful to state Fano's lemma in the following way.

**Lemma 1.** *Let $X$ be uniformly distributed over $[N]$. Then for any random variable $Y$ and any function $f$,*

$$\mathrm{P}(f(Y) \neq X) \geq 1 - \frac{\ln 2 + I(X; Y)}{\ln N}.$$

*Proof.* A usual statement of Fano's lemma is that, if there exists some function $f$ satisfying $f(Y) = X$ with probability $p$, then

$$H(X \mid Y) \leq h(p) + (1 - p)\ln N,$$

where $h(p)$ is the binary entropy function. Since $h(p) \leq \ln 2$ and $H(X) = \ln N$, this implies that

$$\mathrm{P}(f(Y) = X) = p \leq 1 - \frac{H(X \mid Y) - \ln 2}{\ln N} = \frac{\ln 2 + \ln N - H(X \mid Y)}{\ln N} = \frac{\ln 2 + I(X; Y)}{\ln N},$$

which gives our statement above. $\square$

Now we proceed to the proof of Proposition 4, restated for convenience.

**Proposition.** *Let $C > 0$ be arbitrary. Over any regime where*

$$d \leq Ck \ln N, \quad \omega(1) \leq k < N/2,$$

*for Rademacher dictionaries it holds that*

$$\liminf_{N \to \infty} b(d, k, N; \text{top-}k) \geq 1 - \frac{C}{2}.$$

This is a consequence of the following explicit bound on the failure probability $b(d, k, N; \text{top-}k)$.

**Lemma 2.** *For all integers $d \geq 1, k \geq 1$ and $N \geq 1$,[4]*

$$b(d, k+1, N+k; \textit{top-}k) \geq 1 - \frac{\ln 2}{\ln N} - \frac{d}{2k \ln N}(1 + O(k^{-1})).$$

Indeed, given any $k \geq 2$ and $N \geq k$, invoking Lemma 2 with the change of variables

$$k' = k - 1, \quad N' = N - k + 1$$

gives a lower bound on $b(d, k, N; \text{top-}k)$:

$$b(d, k, N; \text{top-}k) = b(d, k'+1, N'+k'; \text{top-}k)$$
$$\geq 1 - \frac{\ln 2}{\ln(N - k + 1)} - \frac{d(1 + O(k^{-1}))}{2(k-1)\ln(N - k + 1)}.$$

Under the regime $d \leq Ck \ln N$ and $\omega(1) \leq k \leq N/2$,

$$\frac{d(1 + O(k^{-1}))}{2(k-1)\ln(N - k + 1)} = \frac{d}{2k \ln N} \frac{1 + O(k^{-1})}{1 + o(1)} \leq \frac{C}{2} + o(1)$$

and of course

$$\frac{\ln 2}{\ln(N - k + 1)} = o(1).$$

We conclude overall that

$$b(d, k, N, \text{top-}k) \geq 1 - \frac{C}{2} + o(1),$$

which proves our result. We proceed to prove Lemma 2.

*Proof.* Define three independent random variables:

- Let $F \in \mathbb{R}^{d \times (N+k)}$ be a Rademacher dictionary;

- Let $S$ be uniformly random in $[N]$;

- Let $\sigma$ be a random permutation of $[N + k]$.

Define the vector

$$X_i = \begin{cases} 1 : i = S \vee i > N \\ 0 : \text{otherwise.} \end{cases}$$

We write $(F^0, F^1)$ for the first $N$ and last $k$ coordinates of $F$, and similarly write $(X^0, X^1)$ for the first $N$ and last $k$ coordinates of $X$. Thus, the superposition code for $X$ under $F$ can be written as

$$Y = FX = F^0 X^0 + F^1 X^1.$$

The vector $F^0 X^0 = F_S$ is the $S$th codeword of $F$, and $F^1 X^1$ is a superposition of the last $k$ codewords of $F$. We will write $\sigma(X)$ and $\sigma(F)$ for the permutations of $X$ and $F$ under $\sigma$ defined in the natural way. Under these conditions, $(\sigma(X), \sigma(F))$ is an independent pair of a uniformly random $(k+1)$-sparse vector and a Rademacher dictionary.

Now, define a function $g: \mathbb{R}^{d \times N} \times \mathbb{R}^d \to [N]$ by

$$g(F^0, y) = \arg\max_{i \in [N]} \langle F_i^0, y \rangle,$$

---

[4]Note that we refer to the top-$k$ algorithm, although the number of selected codewords (formerly $k$) is $k + 1$.

breaking ties arbitrarily. We claim that $g(F^0, Y) = S$ with at least the same probability that the top-$k$ algorithm recovers $\sigma(X)$ from the input $(Y, \sigma(F))$. Indeed, if top-$k$ succeeds, then it must be the case that

$$\langle F_S, Y \rangle \geq \langle F_i, Y \rangle$$

for all $i \in [N]$, because otherwise top-$k$ would select an index in the set $\sigma([N] \setminus \{S\})$, while $\sigma(X)$ is zero at these indices. Under these conditions, it is clear that $g(F^0, Y) = S$ with average probability no smaller than the probability that top-$k$ succeeds. This proves that

$$b(d, k+1, N+k; \text{top-}k) \geq \text{P}(g(F^0, FX) \neq S).$$

Now we will lower bound the probability $\text{P}(g(F^0, FX) \neq S)$ by combining Fano's inequality (in the form of Lemma 1) with our upper bound on mutual information (Proposition 8).

Applying Lemma 1 to the map $g$, we have

$$\text{P}(g(F^0, FX) \neq S) \geq 1 - \frac{\ln 2}{\ln N} - \frac{I(S; (F^0, FX))}{\ln N}. \tag{9}$$

Since $F^0$ is independent from $S$, $I(S; (F^0, FX)) = I(S; FX \mid F^0)$. Furthermore, because the conditional distribution of $FX$ given $S$ and $F^0$ depends only on $F_S$, $I(S; FX \mid F^0) = I(F_S; FX \mid F^0)$. All together,

$$I(S; (F^0, FX)) = I(F_S; FX \mid F^0).$$

Conditional on $F^0$, we can apply Proposition 8 to the pair $(F_S, FX)$, since $F_S \in \{-1/\sqrt{d}, 1/\sqrt{d}\}^d$ and $FX = F_S + F^1 X^1$ is a sum of $F_S$ with $k$ independent Rademacher vectors. Since the bound in question is independent of $F^0$, we conclude overall

$$I(F_S; FX \mid F^0) \leq d \left( \frac{1}{2k} + O(k^{-2}) \right).$$

Combining Equation (9) with this last inequality gives

$$\begin{aligned} b(d, k+1, N+k; \text{top-}k) &\geq \text{P}(g(F^0, FX) \neq S) \\ &\geq 1 - \frac{\ln 2}{\ln N} - \frac{d}{\ln N} \left( \frac{1}{2k} + O(k^{-2}) \right) \\ &= 1 - \frac{\ln 2}{\ln N} - \frac{d}{2k \ln N} (1 + O(k^{-1})) \end{aligned}$$

as desired. $\qquad \square$

## F    Proof of Proposition 5

Now we turn to the proof of Proposition 5, restated below. Recall that $b(d, k, N; \tau)$ is the failure probability for threshold decoding at level $\tau$.

**Proposition.** *For $\eta \in (0, 1]$, let $k \leq N^\eta$. Then for any $\epsilon > 0$, over the regime*

$$d \geq (1 + \epsilon) 2(1 + \sqrt{\eta})^2 k \ln N, \quad \tau \in \left[ \frac{1}{2} + (1 - \eta) \frac{k \ln N}{d}, (1 + \sqrt{\eta})^{-1} \right],$$

*for Rademacher dictionaries it holds that*

$$\lim_{N \to \infty} b(d, k, N; \tau) = 0.$$

Our proof relies on the following bound for $b(d, k, N; \tau)$, valid for Rademacher dictionaries.

**Lemma 3.** *For all $d, k, N \in \mathbb{N}$ and $\tau \in (0,1)$,*

$$b(d, k, N; \tau) \le k \exp\left(-\frac{(1-\tau)^2 d}{2k}\right) + (N-k) \exp\left(-\frac{\tau^2 d}{2k}\right).$$

*Proof.* Suppose w.l.o.g. that $X = \{1, \ldots, k\}$ and let $Y = FX = F_1 + \cdots + F_k$.

Define the families of events

$$
\begin{aligned}
A_i &= [\langle F_i, Y \rangle < \tau] \quad \text{for } 1 \le i \le k, \\
B_i &= [\langle F_i, Y \rangle \ge \tau] \quad \text{for } i > k.
\end{aligned}
$$

Let $R_t$ be a sum of $t$ independent Rademacher variables, as in Proposition 7. For $i > k$, $\langle F_i, Y \rangle$ is distributed as $R_{kd}/d$. Thus, by a Chernoff bound,

$$\mathrm{P}(B_i) = \mathrm{P}(\langle F_i, Y \rangle \ge \tau) = \mathrm{P}(R_{kd} \ge d\tau) \le \exp\left(-\frac{\tau^2 d}{2k}\right).$$

Similarly, for $i \le k$, $\langle F_i, Y \rangle = \langle F_i, F_i \rangle + \langle F_i, \sum_{j \ne i} F_j \rangle$ is distributed like $1 + R_{(k-1)d}/d$, and so

$$
\begin{aligned}
\mathrm{P}(A_i) = \mathrm{P}(\langle F_i, Y \rangle < \tau) &= \mathrm{P}(R_{(k-1)d} < (\tau - 1)d) \\
&\le \mathrm{P}(R_{(k-1)d} \ge (1 - \tau)d) \le \exp\left(-\frac{(1-\tau)^2 d}{2k}\right).
\end{aligned}
$$

Our conclusion follows from a union bound

$$b(d, k, N; \tau) = \mathrm{P}\left(\bigcup_i A_i \cup \bigcup_i B_i\right) \le \sum_{i=1}^k \mathrm{P}(A_i) + \sum_{i=k+1}^N \mathrm{P}(B_i).$$

$\square$

The proof is now a matter of calculation.

*Proof of Proposition 5.* Denote $C = d/(k \ln N)$. Together, the assumption $k \le N^\eta$ and Lemma 3 let us bound $b(d, k, N; \tau)$ by a sum of powers of $N$ with exponents depending only on $(\eta, C, \tau)$:

$$b(d, k, N; \tau) \le N^\eta \exp\left(-\frac{(1-\tau)^2 C k \ln N}{2k}\right) + N \exp\left(-\frac{\tau^2 C k \ln N}{2k}\right) = N^{P_0} + N^{P_1}$$

where

$$P_0 = \eta - \frac{(1-\tau)^2 C}{2}, \quad P_1 = 1 - \frac{\tau^2 C}{2}.$$

Denote $\tau_0(C) = \frac{1}{2} + (1 - \eta)C^{-1}$ and $\tau_1 = (1 + \sqrt{\eta})^{-1}$ for the lower and upper bounds on $\tau$ in our statement, and $C_0 = 2(1 + \sqrt{\eta})^2$. Note that when $C \ge C_0$ we indeed have

$$\tau_0(C) \le \tau_0(C_0) = \frac{1}{2} + \frac{1 - \eta}{2(1 + \sqrt{\eta})^2} = (1 + \sqrt{\eta})^{-1} = \tau_1.$$

We will show that both $P_0$ and $P_1$ are at most $-\epsilon\eta$ under our conditions on $C$ and $\tau$, which shows overall that $b(d, k, N; \tau) = O(N^{-\epsilon\eta})$.

*Bounding $P_0$.* Since $P_0$ is increasing in $\tau$ and decreasing in $C$, for $\tau \le \tau_1$ and $C \ge (1 + \epsilon)C_0$ we have

$$P_0 \le \eta - \frac{(1 - \tau_1)^2 (1 + \epsilon) C_0}{2} = \eta - \frac{\eta}{(1 + \sqrt{\eta})^2} \cdot \frac{(1 + \epsilon) \cdot 2(1 + \sqrt{\eta})^2}{2} = \eta - (1 + \epsilon)\eta = -\epsilon\eta.$$

*Bounding $P_1$.* Setting $P_0 = P_1$ and solving for $\tau$ gives $\tau = \frac{1}{2} + (1 - \eta)/C = \tau_0(C)$, so $P_0$ and $P_1$ are equal at the lower endpoint of our interval. Since $P_1$ is decreasing in $\tau$, for $\tau \ge \tau_0(C)$ we have

$$P_1 \le P_1(\tau_0(C), C) = P_0(\tau_0(C), C) \le -\epsilon\eta$$

where the last inequality follows from our bound on $P_0$, using the fact that $\tau_0(C) \le \tau_1$. $\square$

# G   Numerical Results on Spherical Dictionaries

Figure 5 provides experimental results identical to Figure 3 except for the use of spherical dictionaries instead of Rademacher dictionaries. Specifically, we sample codewords $F_i$ as independent vectors uniformly distributed over the unit sphere. In our experiments, coeffients are represented by half-precision floating points. Figure 5 and Figure 4 are almost visually identical. The small differences that exist are most visible for small values of $k$. (Small values of $k$ correspond to "buckets" in the range of $\eta$.)

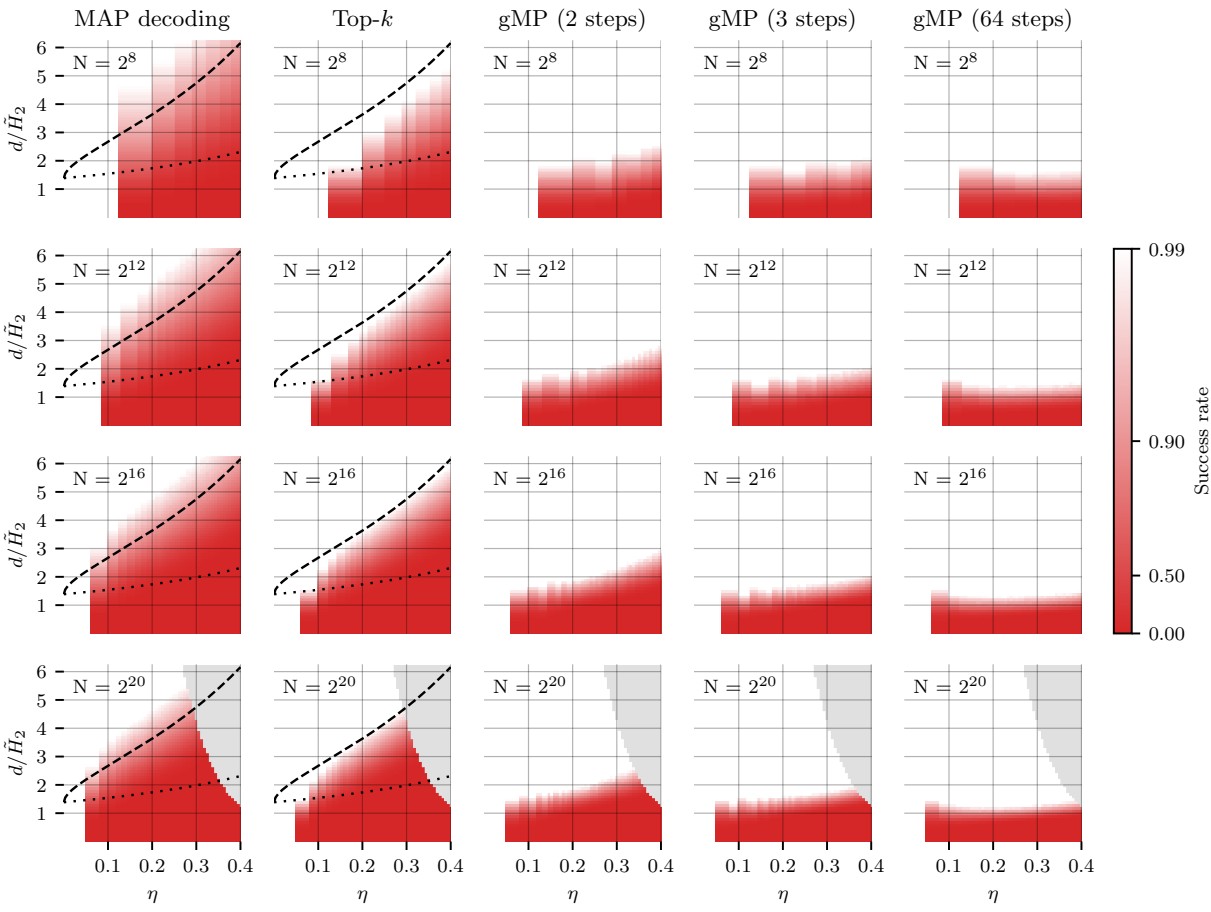

Figure 5: Empirical results on spherical dictionaries, otherwise with the same experimental setup described in Section 4.2. Compare to Figure 3.

