# OpenReview forum: "How Much Information Fits in a Vector?"
_TMLR — Decision pending for TMLR_

### Review · Reviewer_yrCr · 2026-03-31

**Summary Of Contributions:**

This paper aims to provide theoretical approximations of the probability of successfully decoding, with various decoding strategies, the original features by only seeing superposition of $k$ of those features obtained from some one-step map $F$. That is, a vector $\boldsymbol{x} \in \{0, 1\}^N$ of $N$ binary features is mapped to a superposition $\boldsymbol{y} = F \boldsymbol{x}$ by a matrix $F \in \mathrm{R}^{d \times N}$. This problem is studied through a toy example where the columns of $F$ consists of randomly sampled vectors, either Rademacher or spherical, for the decoding strategies of threshold decoding, top-$k$ and GMP. Specifically, the authors derive an asymptotic lower bound on the encoding dimension $d$, which depends on the sparsity level $k$ and $N$, for it to be possible for top-$k$ to decode a superposition with some probability. They also derive a similar asymptotic lower bound on $d$ for threshold decoding to decode successfully almost surely given a fitting threshold. The authors then continue with empirical observations of successful decoding and compare it to the derived bounds.

**Additional Comments:**

Additional remarks.
1. The discussion below corollary 1 compares incomparable quantities. One is a upper bound that holds for spherical dictionaries and top-$k$ decoding and the other is an lower bound for Rademacher dictionaries and threshold decoding. A priori, these have nothing to do with each other, and so the provided discussion is moot.
2. Algorithm 3, "choose any step sizes [...] and $\max_t k_t$ is minimised", is there not a unique choice of step sizes; pick each $k_i = k / T$, otherwise how can $\max_t k_t$ be minimised?
3. Put appropriate references to App. B and C in the main text, so that the reader does not have to guess at jumps in derivations.
4. A direct example of $\boldsymbol{x}$, $\boldsymbol{y}$ and $F$ in the introduction would help the clarity of the exposition, e.g. $\boldsymbol{x}$ are characteristic features of birds, $F$ maps to a latent vector $\boldsymbol{y}$ of more abstract features.

#### Feedback
The authors clearly have a good handle on the necessary theory and I appreciate at least the aim of being mathematically formal, yet that only makes it more disappointing to then see sloppy explanations, proofs and empirical observations. The paper, right now, is in an awkward middle ground of potentially having some truly informative insights, but without the appropriate support. Bold, and potentially overly strong, claims are made to connect to recent efforts in SAEs and language models, but these are also not supported enough. At least an empirical study on the success rate of actual SAEs would be necessary for that, together with a thorough rework of the paper's argumentation. I strongly encourage the authors to restructure the paper, make it more formal, and align its claims for a future resubmission.

**Audience:**

Yes

**Audience Explanation:**

Some interesting behaviour is definitely present, but the paper does not provide substantial evidence for it. If it could fix its lack of rigour and everything still holds, then the paper could provide nice insights.

**Broader Impact Concerns:**

None.

**Claims And Evidence:**

No

**Claims Explanation:**

Contribution 1 states that one-step estimates used by sparse auto-encoders can reliably decode superpositions under certain assumptions on $d$ and that the provided bounds can accurately predict the empirical performance of top-$k$ decoding. I have the following three counter arguments to this claim.

**Counter argument 1.** The authors only theoretically show the condition under which threshold decoding can reliably decode **superpositions of certain random vectors**. The argument in section 5 is meant to provide some connection to the superpositions encountered in SAEs, but it is not a direct connection and there is no empirical evidence provided either (no empirical experiment with SAEs is shown, nor with any other neural model). Additionally, the authors do not show conditions under which  top-$k$ can **reliably** decode; they only provide a regime in which top-$k$ has some probability of not failing, but not that it will succeed asymptotically. The same lack of empirical evidence with SAEs mentioned before for threshold decoding holds for top-$k$ as well. Hence, the connection to SAEs is purely speculative.

**Counter argument 2.** All empirical evidence of the provided bounds with respect to predicting the success rate/sample efficiency of decoding methods is questionable. Only a gradient graph is provided with a (two) piece-wise linear gradient going from 0 to 1 with middle value 0.9, without any explanation as to why a specific value of 0.9 success rate would be what the lower bound tracks. It is hard to disentangle the specific choice of gradient from the actual tracking behaviour. If either a purely logarithmic gradient or purely linear gradient would be shown, then at least the visual indication would be more interpretable. Apart from that, even the visual tracking behaviour of lower bound and empirical evidence for the sublinear sparsity case (Figure 4) is not always convincing; top-$k$ bounds are often much away from the (arbitrary) transition at 0.9 success rate.

**Counter argument 3.** The overall presentation, both formal and informal is not rigorous. For one, the authors often use speculative terms as "naturally we have" (above proposition 3), "similar results can naturally be proven" (below proposition 3), "it is likely possible" (below proposition 3), "A simple extension would prove" (below proposition 4), etc. instead of providing hard, scientific result of these statements. While I am not saying every single step needs elucidation, the paper is riddled with speculations and so-called "simple or trivial" statements for results that the authors actually need. It is not scientific. Science does not proclaim, it explains. Either provide evidence, or do not mention it at all. Similar non-rigorous statements and derivations also undermine the proofs of proposition 3 and 4. For the proof of proposition 3 in App. D, I have two potential failures.
1. "It is easy to show that [...]. Let us condition on this event." The statement does not seem easy to show to me, and as part of a proof requires a formal argument. It is also unclear what ramifications there are from "conditioning on this event ($k > 2$)".
2. "[...] the pair $(F_{i}, \frac{Y}{||Y||})$ is a pair of independent, **uniform** draws from the unit sphere, [...]" I do not see how $\frac{Y}{||Y||}$ would be a **uniform** random variable on the unit sphere. Is the **normalised** sum of uniform unit sphere random variables again uniform on the unit sphere?

For the proof of proposition 4 in App. D, should it not be that $\braket{F_{i}, Y}$ is distributed according to $1 + R_{\textcolor{red}{(n - k - 1)d}} /d$? No clear explanation is given why, also for the other case, the distributions of $\braket{F_{i}, Y}$ are as stated for the two cases.

**Requested Changes:**

Please address the three counter arguments raised in the above paragraphs. Additional smaller remarks can be found below.

---

> ### Author Response · Authors · 2026-04-21
>
> Thank you for your careful feedback. We appreciate the demand for precision and agree that several cases in our work should be amended. The most serious of these, by far, is Proposition 3. As you have noted, the proof depends on a subtle claim that is not justified. The proof strategy also has a more serious flaw. However, we have recently succeeded in proving a slightly altered version of Proposition 3 using a different proof strategy, which proves exactly the same asymptotic lower bound on sample complexity for top-$k$ that we had claimed previously. We detail the situation below and in our reply to Reviewer YDG8. However, the other necessary revisions seem to be relatively minor.
>
> ### Reply to counter argument 1
> We certainly do not mean to claim that our guarantees are valid outside of the toy setting we have described. We intended for Contribution 1 to be understood in the context of the previous paragraph, where we note that our analysis focuses on a toy scenario. Indeed, it is not possible to prove any kind of guarantee without some assumption on the dictionary. However it may be appropriate to add a clarifying phrase to Contribution 1, like "in our toy scenario."
>
> Concerning our guarantees for top-$k$, Proposition 4 does in particular provide a condition for asymptotic success of top-$k$ decoding, since the failure probability of top-$k$ can never be larger than the failure probability of threshold decoding. However we realize it would be useful to summarize these results somewhat more conveniently. Overall, given our amendment to Proposition 3 (see below), we will add a corollary stating that, where $C(\eta; m)$ is the infimum of all constants such that the failure probability of a method $m$ converges to $0$ over the regime $k \sim N^\eta$ and $d \ge C \ln \binom N k,$ we have for Rademacher dictionaries that
> $$
> \frac{2}{1 - \eta} \le C(\eta; \text{top-}k) \le C(\eta; \text{MAP}) \le
> \frac{2(1 + \sqrt \eta)}{1 - \sqrt \eta}.
> $$
>
> On the speculative nature of the connection between our results and applications of SAEs, we are in full agreement and have tried to make this clear throughout our work.
>
> ### Reply to counter argument 2
> The choice of $0.9$ failure probability as a breakpoint is indeed arbitrary, but this does not diminish the evidence provided by our graphs. The "lower bounds" in question are related to the asymptotic upper bound on sample complexity given by Proposition 4, and should not be expected to track any particular failure probability. Instead, the expected behavior is that, when $d$ or $d/H$ is a constant factor larger than the value of these expressions, failure probability converges to $0$ as $N \to \infty.$
>
> Our figures support this conclusion and show, moreover, that even for finite values of $N$ we can expect relatively low failure probability as $d$ or $d/H$ exceeds these values by a constant factor. This confirms the prediction of Proposition 4. In Figure 4, we agree that the empirical transition for top-$k$ appears to be slightly earlier than the transition for threshold decoding, but this does not undermine Proposition 4, which is only a sufficient condition.
>
> However, we agree that it could be more understandable to use a sequential color scale measuring the log probability of success. We have made this change to our working draft, and the visual interpretations of both Figure 3 and Figure 4 are unchanged.

---

> ### Author Response · Authors · 2026-04-21
>
> ### Reply to counter argument 3
> In several cases we disagree that our writing is overly informal.
>
> (1) The phrase "naturally we have" above Proposition 3 introduces the fact that the failure probability of top-$k$ is bounded above by the failure probability of threshold decoding, which is justified by the next sentence. A similar observation is made at the end of section 2.1, so it should not come as a surprise to the reader that top-$k$ is uniformly better than threshold decoding.
>
> (2) In the proof of Proposition 3, we use the fact that the normalized sum of independent random draws from a unit sphere is again uniform over the sphere, which is intuitive and arguably the kind of result that may be asserted without proof. One way to prove it is the following. Let $Q$ be any linear isometry. Then where $(F_1, \dots, F_k)$ are independent and uniformly random on the unit sphere, $(Q F_1, \ldots, Q F_k)$ has the same distribution as $(F_1, \dots, F_k).$ It follows that
>  $$
>  (Q F_1 + \dots + Q F_k) / \lVert Q F_1 + \dots + Q F_k \rVert = Q (F_1 + \dots + F_k)/\lVert F_1 + \dots F_k \rVert
> $$
> has the same distribution as $(F_1 + \dots + F_k)/\Vert F_1 + \dots + F_k \rVert.$ But a distribution is uniform over the unit sphere if and only if it is supported on the sphere and is invariant under all isometries.
>
> (3) In the proof of Proposition 4, the quantity $\left\langle  F_i, \sum_{j \neq i} F_j  \right\rangle$ expands into a sum of $(k - 1) d$ products of pairs of coefficients, because each vector has $d$ dimensions and the sum has $(k - 1)$ terms. These products are statistically independent, and so the inner product is distributed like a multiple of $R_{(k - 1)d}.$
>
> (4) "A simple extension would prove" below Proposition 4 is arguably justified because the referred discussion in Appendix D explicitly shows how the MAP threshold relates to the proof of Proposition 4, and from this relationship it follows that putting $\tau$ equal to the MAP threshold would give the same conclusion as Proposition 4.
>
> However, we are thankful for your careful reading, and agree that several situations can be revised, as follows.
>
> (1) First, we ultimately agree that the relationship between the MAP threshold and Proposition 4 could be more explicit. We propose to modify the condition of Proposition 4 to
> $$
> d \ge (1 + \epsilon) 2(1 + \sqrt \eta)^2 k \ln N, \quad \tau \in \left [\frac 1 2 + (1 - \eta) \frac{k \ln N} d, (1 + \sqrt \eta)^{-1} \right],
> $$
> which holds with the same conclusion. The proof remains almost the same. Furthermore, we should clarify that the MAP threshold should be defined as $1/2 + (1 - \ln k / \ln N) k \ln N / d.$ This is the formula we used when computing MAP thresholds in our numerical experiments. The current expression, described at the start of page 7, includes additional terms of order $O(k/N).$
>
> (2) Second, we propose an amendment to Proposition 3. The current proof of this result, besides depending on the subtle claim that you have mentioned, has a serious deficiency: the current argument can only prove that the probability of failure does not converge to $0$ over a certain regime, but cannot prove that it converges to $1.$
>
> We have recently realized how this can be fixed, and have been able to prove a slightly different (but very similar) statement using a different strategy. The situation is described in our reply to Reviewer YDG8. Our proof is essentially a careful formalization of the informal argument described at the end of Section 3.
>
> The new statement holds for Rademacher dictionaries rather than spherical dictionaries, and therefore has the advantage of complementing Proposition 4.

---

> ### Author Response · Authors · 2026-04-21
>
> ### Reply to additional remarks
> (1) It is true that the failure probabilities in Propositions 3 and 4 are incomparable in the sense that our analysis does not rigorously show one is larger than the other, but the discussion in question is simply an honest comparison these two results. Fortunately, with our amendment to Proposition 3, this slightly awkward situation is rectified, and these quantities become properly comparable.
>
> (2) When $k$ is not a multiple of $T$, the goal of this definition is just to assign any step sizes that divide $k$ as evenly as possible. Maybe it would be preferable to simply describe the goal in conversational English, e.g., "choose step sizes $k_i$ evenly such that $k_1 + \dots + k_T = k$"?
>
> (3) We will add a reference to Appendix B on page 10 to justify the bound $\ln \binom N k \le k \ln(e N/k).$ Appendix C, however, seems to only be needed to prove Proposition 1 and state a Chernoff bound used in Appendix D.
>
> (4) The figure on the first page gives a classic example of how a "feature" can be encoded by way of more abstract features. In our example a feature is a position of a point on the plane, and the abstract features correspond to receptive fields. However we could make this connection more explicit by explaining that a set of points could be represented by the linear combination of their "abstract feature vectors" (codewords). In this example, the receptive fields of Figure 1 would correspond to the rows of the matrix $F.$

---

> ### Comment · Action_Editor_mPpQ · 2026-05-24
>
> Dear Reviewer,
>
> We haven't heard much from you, could you please read the rebuttal, engage with the authors if appropriate?
> The authors have also uploaded a modified version, would you mind checking if you believe it to be more rigorous than the original?
>
> It seems you are the only reviewer with concerns regarding correctness, which is a serious issue that should be handled before making a final decision on the manuscript.
>
> Best regards,
>
> AE

---

### Review · Reviewer_YpXg · 2026-04-06

**Summary Of Contributions:**

The authors analyze the superposition code recovery problem from an information-theoretical perspective by establishing bound on the dimension required for successful recovery for "one-step" decoding methods like MAP thresholding and top-$k$. In particular, Proposition 3 provides a lower bound on the required dimension for successful recovery probability of top-$k$, whereas Proposition 4 provides an upper bound on the required dimension for successful recovery probability of thresholding.

**Audience:**

Yes

**Audience Explanation:**

The paper studies "one-step" decoders commonly used in SAEs and uses scales motivated by SAE practice such as $N \approx 2^{20}$ and $k \approx 64$, so it should interest at least some TMLR's audience. While I think this is a solid paper in high dimensional statistics, the authors said "the main goal of this work is to provide a useful analytical starting point for research in interpretability", and I'm not entirely convinced. I'm okay with the random dictionary assumption, but the entire setting assumes there is a ground-truth $f$ to be recovered, and there is no such ground truth in LLM latent representations, so it's unclear to me what "successful recovery" mean in this context.

**Claims And Evidence:**

Yes

**Claims Explanation:**

I didn't check the proof in the appendix carefully, but Proposition 3 and 4 seem mathematically rigorous. They also matches the empirical experiments in Figure 3 and Figure 4. Moreover, the lower bound in Proposition 3 is only off by a multiplicative constant $(1+\sqrt{\eta})^2$ from the upper bound in Proposition 4, which means both are right (these two bound are comparable because top-$k$ is better than thresholding).

**Requested Changes:**

1. Provide an intuitive explanation for the extra $(1+\sqrt{\eta})^2$ in Proposition 4.
1. In Figure 3, it's perhaps useful to plot the lower bound in Proposition 3 as well, to see how optimistic it is for MAP thresholding (which probably has a higher lower bound than top-$k$). The same goes for Figure 4.
1. I wish the authors can elaborate more on how this result is relevant for LLM latent representations. For example,
    1. Proposition 4 focus on sublinear sparsity, but it's unclear to me why this is a good mental modal (after all, $k$ and $N$ are both constants), while most compressive sensing literature are happy with linear sparsity.
    1. Regarding "The most obvious result of our findings is to suggest that the performance of SAEs may be improved by using slightly more computation to solve the sparse reconstruction problem." Can you be more concrete about which performance aspect will be improved, e.g., which metric are we talking about here?
1. While the paper provides a solid theoretical grounding for "one-step" decoding methods like MAP thresholding and top-$k$, the analysis for generalized matching pursuit (GMP) is purely numerical. Can you comment on what is the intuition behind the superior statistical accuracy of GMP? Is that related to the fact that GMP is a better optimization algorithm, which leads to a lower "empirical risk", or are there deeper reasons?
    1. It would be really cool if you can connect "how good a decoder is at solving the optimization problem", i.e., reconstruction quality, and "how many dimensions per nat is required for successful recovery", but I guess that's a topic for another paper.

---

> ### Author Response · Authors · 2026-04-21
>
> Thank you for your feedback and detailed requests. Replying in order:
>
> (1) To understand this factor, it may help to realize that the lower bound of $2/(1 - \eta)$ can be understood, roughly, as the critical value such that $d \ge (1 + \epsilon) C H$ guarantees that just one feature of $X$ can be accurately recovered if we treat the remaining $(k - 1)$ features as noise. (In fact, this idea is formalized in our new proof for Proposition 3.) Thus, the additional factor of $(1 + \sqrt{\eta})^2$ can be understood as the constant factor increase of $d$ needed for this restricted event to hold with high enough probability that _every_ feature is recovered with high probability.
>
> (2) We agree, and have improved Figure 4 by indicating both the upper and lower critical values of $2(1 + \sqrt \eta)/(1 - \sqrt \eta)$ and $2 / (1 -\eta).$ This shows that the latter is relatively pessimistic in practice for all but the smallest values of $\eta.$
>
> (3.1) As we have clarified in our reply to reviewer YDG8, in the linear sparsity regime $k \ge \epsilon N$ reconstruction is only possible so long as the dictionary $F$ has bounded "aspect ratio," meaning $d/N \ge \delta.$ For instance, see Figure 2 of [1]. A representative result would be that, for $d/N \ge 0.1,$ a linear program can recover $k$-sparse vectors satisfying $k/N \le 0.015.$ (This is obtained from Figure 2 of [1] with $\delta = 0.1$ and $\rho = 0.15.$)
>
> In practice, the parameters $(k, d, N)$ are indeed constants, but for sparse autoencoders we intuitively expect to have $d/N$ vanishingly small; for language models we have $d \approx 1000$ but $N$ extremely large, equal to the size of some collection of "concepts". Thus, results like those illustrated in [1] should not be informative for sparse autoencoders.
>
> We also highlight that our work provides one characterization of the sublinear assumption: we show that it is exactly in sublinear regimes that threshold decoding and top-$k$ can avoid being information-theoretically inefficient, in the sense of only requiring a finite bound on $d/H.$ To our knowledge this is an original result, and may be useful when reasoning about LLM latent representations.
>
> (3.2) Our suggestion is that, by using an algorithm like GMP, it may be possible for the decoder to read latent features much more reliably when provided with the correct dictionary $F.$ Therefore, if this hypothesis holds, using GMP both at dictionary-learning time and inference time could decrease metrics like degradation in downstream performance when activations are replaced with SAE reconstructions.
>
> (4) Yes, we can suggest some intuition some for why an algorithm like GMP improves over an algorithm like top-$k$, which we can include in our revisions. For instance, the following explanation can accompany our definition of the GMP algorithm.
>
> One point of view, commonly used to motivate methods like ISTA and LISTA (also described in [2]), is that iterative methods let estimates for the different latent quantities $X_i$ compete for expression. For instance, suppose a set $S$ of codewords have relatively large inner product with each other but only one belongs to a given code $Y.$ With top-$k$ decoding, it is possible that multiple elements from $S$ would make it into the $k$ codewords with largest inner product. However, GMP would ideally select the top codeword in its first iteration. Since this codeword is subtracted from the remainder in subsequent iterations, the remaining elements of $S$ would be less likely to be selected moving forward.
>
> [1] Donoho, David L., and Jared Tanner. “Precise Undersampling Theorems.”
> [2] Gregor, Karol, and Yann LeCun. “Learning Fast Approximations of Sparse Coding.”

---

> ### Author Response · Authors · 2026-05-04
>
> We'd also like to defend our claim that this work provides "a useful analytical starting point for research in interpretability."
>
> We agree that the application of our work is not straightforward, in the sense that our theoretical results do not have direct practical applications. Indeed, as you remark, it is unlikely that the "ground truth" model for an activation vector within an LLM is anything like what we have described in our toy example. However, our results provide some intuition that may extend to the general problem of decoding representations that involve superposition.
>
> Specifically, suppose an activation vector can be interpreted as a sum $Y + Z,$ where $Y$ is a representation for some isolated feature and $Z$ is a superposition of representations for other features. Based on relatively well-known ideas from statistics, we can make two observations about the problem of inferring $Y$:
>
> 1) When the scale of $Y$ is sufficiently large compared to the scale of $Z,$ it may be feasible to estimate $Y$ by treating $Z$ as Gaussian noise. In our toy example, this strategy results in the method of threshold decoding.
>
> 2) However, it may be possible to infer $Y$ much more reliably by simultaneously solving an inference problem for $Z.$ For instance, results on compressive sensing in a linear regime show that it is possible to infer the latent coefficients of a superposition code even when the relative scale of the isolated features $X_i F_i$ is vanishingly small. In this situation, a method like threshold decoding will fail catastrophically.
>
> In practice, when we seek to "decode" activation vectors using a sparse autoencoder, it is not clear whether the subproblem of sparse reconstruction should be solved by a "one-step method" or by some more elaborate inference method. Unfortunately, the gap between one-step methods and compressive sensing algorithms is not well-studied. Our findings suggest that, one-step methods do not fail catastrophically in settings of extreme sparsity, but relatively cheap extensions of these methods may be able to read significantly more information.
>
> These comments could be integrated in our conclusion.

---

### Review · Reviewer_YDG8 · 2026-04-18

**Summary Of Contributions:**

This paper focuses on the sparsity recovery of the sparse coding problem. That is, to retrieve the high-dimensional latent representation from an activation vector. It studies the theoretical limits for threshold decoding and top-k decoding under random spherical dictionaries or the Rademacher dictionary. This theoretical limit aligns well with the simulation studies. It also shows that GMP (Generalized Match Pursuit) leads to significant performance improvement.

The theoretical analysis: in Appendix Section D, proof of Proposition 3: It only shows that $P(B| ||Y||\geq \sqrt k, F_1^TY<1)\rightarrow 1$, but doesn't show that $P(||Y||\geq \sqrt k, F_1^TY<1)\rightarrow 1$.

The main implication of this paper is that $d/H>C(\eta)$ is a sufficient condition for sparse recovery. If I view the problem as a linear regression, then the standard result gives a sufficient condition of $d/[k\log(N/k)]$ being sufficiently large. Therefore, the contribution is quite subtle. And the impact of $\eta$ is only theoretically studied for the threshold decoding method. Therefore, the paper doesn't quite answer its title, "How Much Information Fits in a Vector?"

**Additional Comments:**

None

**Audience:**

Yes

**Audience Explanation:**

While the paper's motivation is related to deep learning, the result may not have a direct impact on the practitioners of deep learning. But  ML theorists should be interested in such an analysis.

**Claims And Evidence:**

No

**Claims Explanation:**

The claims made by the paper are either justified by rigorous analysis or empirical experiments. My major concern is that $d/H\geq C(\eta)$ for some increasing function $C$ is only a sufficient condition, and maybe misleading.
For example, if I consider a random Gaussian dictionary, i.e., all entries of $f_i$ are iid normal. Then, in the view of linear regression $y=Fx$, the minimax L2 rate is $\sqrt {(2+o(1))k\log(N/k)/d}$ (refer to "SLOPE is Adaptive to Unknown Sparsity and Asymptotically Minimax"). Since all entries of $x$ is either 0 or 1, we can ensure sparsity recovery as long as $\sqrt {(2+o(1))k\log(N/k)/d}>1$, i.e. $d/[k\log(N/k)] \geq 2+o(1)$. The bound I derive here is irrelevant to $\eta$ ( and actually better than (5) in the paper) using the SLOPE estimating procedure.

I suppose the authors' claims are correct only if we consider ''one-step'' estimations.

**Requested Changes:**

I will request that the authors comprehensively compare their results with those in the high-dimensional linear regression literature. More in-depth analysis is necessary to strengthen its argument.

---

> ### Author Response · Authors · 2026-04-21
>
> Thank you for bringing up the connection with existing results on sparse linear regression. Indeed we agree that some clarification should be added on how our own work compares with these results.
>
> Furthermore, we have also noticed an error in the current proof of Proposition 3, which you seem to have mentioned. However, we have been able prove a very slight modification to Proposition 3 using a different technique. With this correction, our paper shows conclusively that our sufficient condition on the ratio $d/H$ is not misleading, in the sense that no bound of the form $d \ge C k \ln(N/k)$ can be sufficient for either threshold decoding or for top-$k$ over all exponents $\eta$. Hopefully this resolves your concern.
>
> In the following, we (1) summarize what should be added to the paper concerning the relationship between our theoretical results and previous work, and (2) briefly describe our proposed amendment to Proposition 3.

---

> ### Author Response · Authors · 2026-04-21
>
> ### (1) Relationship with Prior Work
> We should acknowledge that $d = \Omega( k \ln (N/k))$ dimensions are known to be sufficient for certain relatively tractable decoding methods to succeed at recovery of a sparse $\{0, 1\}$ vector in various moderate situations. In particular, conditions of this form hold both for linear and sublinear sparsity. Besides your suggested reference of [1], another concise reference on this fact that we are aware of is given in [2], by Corollary 7.3.
>
> To give our work some context, it is helpful to mention the particular form that this condition takes in the regimes of linear and sublinear sparsity. When $k \sim \epsilon N$ for $\epsilon \in (0, 1),$ the quantity $k \ln(N/k)$ is linear in $N,$ so the condition $d \ge \Omega(k \ln(N/k))$ amounts to a lower bound on the aspect ratio $r = d/N$ of the dictionary. Note that the entropy of a uniformly random $k$-sparse subset of $[N]$ is $H = \ln \binom k N \sim h(\epsilon) N$ where $h(\epsilon)$ is the binary entropy function, so up to a constant this matches the sample complexity we would expect from information theory. However, it is not an easy problem to determine the minimum value of the ratio $d/N$ that suffices asymptotically for a given value of $\epsilon$ and for a given decoding method. [3] provides some results on these phase transitions.
>
> In the sublinear regime $k \sim N^\eta,$ we have $H = k \ln(N/k) + O(k) = k \ln(N/k) (1 + O(1 / \ln (N/k))) \sim k \ln(N/k),$
> and so $d \ge C k \ln(N/k)$ is asymptotically implied by $d/H \ge C'$ for any $C' > C.$ In particular, if $d = \Omega(k \ln(N/k))$ dimensions are sufficient, then $d \ge C H$ is sufficient for some constant $C$ not depending on $\eta.$ As in the linear sparsity case, it does not appear to be an easy problem to determine critical values of $d/H$. Some recent work ([5], [6]) has investigated the information-theoretic limits of this problem and shown under certain conditions that the maximum likelihood estimator is optimal.
>
> Now, in [1] [2] and [3], decoding is performed by solving certain convex programs. Indeed, to the best of our knowledge, all methods with general sample complexity of $O(k \ln(N/k))$ require executing some iterative algorithm in practice. In the context of sparse autoencoders we consider, the number $N$ of codewords is extremely large and the chosen inference procedure is used as a subroutine to solve the problem of dictionary learning. Therefore, current SAEs use extremely limited methods, comparable to the threshold decoding and top-$k$ algorithms described in our paper. Our work studies the sample complexity of these methods under some strong distributional assumptions. As we have remarked in the paper, the problem does not seem to be well-studied; practically the only reference we could find on the performance of "one-step methods" is [4], which proves an upper bound but no lower bound for sample complexity.
>
> With this background, we can describe the theoretical contribution of our work as follows. Let $C(\eta; m)$ denote the critical value for the ratio $d/H$ for a given method $m$ over the regime $k \sim N^\eta.$ (That is, the infimum of all constants $C$ such that $d / H \ge C$ is a sufficient condition for failure probability to converge to $0$ for large $N.$) Then, with our new statement of Proposition 3, we prove for Rademacher dictionaries that
> $$
> \frac{2}{1 - \eta} \le C(\eta; \text{top-}k) \le C(\eta; \text{MAP}) \le
> \frac{2(1 + \sqrt \eta)}{1 - \sqrt \eta}.
> $$
> Since the lower bound is unbounded as $\eta \to 1,$ the behavior of the upper bound is not misleading; for these two methods, no condition of the form $d \le C k \ln(N/k)$ would be sufficient as $\eta \to 1.$ Furthermore, our upper bound improves strictly on the best comparable condition described in [4], which gives expressions for the sample complexity that do not depend on $\eta.$ Specifically, the best condition of [4] that holds in our scenario is similar to the uniform sufficient condition $d \ge (1 + \epsilon ) 8 k \ln N$ that we describe following Proposition 4.
>
> [1] Su, Weijie, and Emmanuel Candes. “SLOPE Is Adaptive to Unknown Sparsity and Asymptotically Minimax.” arXiv:1503.08393.
> [2] Donoho, David L., and Jared Tanner. “Counting Faces of Randomly Projected Polytopes When the Projection Radically Lowers Dimension.”
> [3] Donoho, David L., and Jared Tanner. “Precise Undersampling Theorems.”
> [4] Bajwa, Waheed U., Robert Calderbank, and Sina Jafarpour. “Why Gabor Frames? Two Fundamental Measures of Coherence and Their Role in Model Selection.”
> [5] Gamarnik, David, and Ilias Zadik. “High-Dimensional Regression with Binary Coefficients. Estimating Squared Error and a Phase Transition.” arXiv:1701.04455.
> [6] Reeves, Galen, Jiaming Xu, and Ilias Zadik. “The All-or-Nothing Phenomenon in Sparse Linear Regression.”

---

> ### Author Response · Authors · 2026-04-21
>
> ### (2) Correction of Proposition 3
> The current proof of Proposition 3 conditions on the event that $\lVert Y \rVert \ge \sqrt{k}$ and $F_1^T Y < 1,$ and shows that in this case that top-$k$ fails with high probability. However, you correctly note that $P(\lVert Y \rVert \ge \sqrt{k} \land F_1^T Y < 1)$ cannot converge to $1,$ and so this proof doesn't support our conclusion that the failure probability of top-$k$ converges to $1$ in a certain regime.
>
> We have been able to prove a slightly different statement using an information-theoretic argument. The new statement is the following:
>
> _Let $C > 0$ be arbitrary. Over any regime where_
> $
> d \le C k \ln N$ and $\omega(1) \le k < N/2,
> $
> _for Rademacher dictionaries it holds that_
> $
> \liminf_{N \to \infty} b(d, k, N; \text{top-$k$}) \ge 1 - \frac C 2.
> $
>
> Our proof is essentially a careful formalization of the informal argument described at the end of Section 3.

---

### Comment · Action_Editor_mPpQ · 2026-04-23

Dear Reviewers,

The authors have submitted their rebuttal and the discussion is underway. Could you please take a look at the authors' responses and verify whether they have successfully defended the value of the paper?

@ reviewer YDG8: many thanks for catching a significant error in the paper. The authors are claiming that it can be fixed by restricting to Rademacher dictionaries. Do you think that a correct proof is credible and that it would still be interesting even with the significant restriction to Rademacher dictionaries? It seems a little counterintuitive to me that this restriction should be necessary. I am not sure if the result can be proved in full generality.


@ Reviewer YpXg, thanks for also catching a lot of sloppy or improper statements. Is the rebuttal helping here?

@authors: I noticed in your answer to reviewer YDG8 that you mention you are able to fix Proposition 3 with "a careful formalization of the informal argument described at the end of Section 3". Based on this, the new proof you allude to cannot be considered peer reviewed. Can you provide a completely **detailed and rigorous** statement and **proof** in markdown exactly as it will appear in the resubmission or camera-ready version, for the reviewers to examine?


Best regards,

AE

---

> ### Author Response · Authors · 2026-04-23
>
> Concerning the fix to Proposition 3, we'd like to clarify that the new statement is not exactly a restriction. In the submitted version, Proposition 3 was stated only for spherical dictionaries, while the modified statement refers only to Rademacher dictionaries. Incidentally, the new proof strategy appears to generalize much more easily to other kinds of dictionaries.
>
> It's our intention to submit a revision of the manuscript, including the new statement and proof, by the end of day today.

---

> ### Author Response · Authors · 2026-04-23
>
> We've uploaded a revised manuscript with changes highlighted in blue. The main revisions are:
>
> (1) Our correction of Proposition 3, including a proof (Appendix D) using a different proof strategy. Again, we'd like to highlight that the new result provides an asymptotic condition on sample complexity that is identical to what we had claimed previously.
>
> (2) A very slight extension to Proposition 4, discussed in our reply to Reviewer yrCr.
>
> (3) A new color scale for Figures 3 and 4, discussed with Reviewer yrCr, and the inclusion of lower bounds suggested by Reviewer YpXg.
>
> (4) An additional clarifying result (Corollary 1), summarizing the results of Propositions 3 and 4 more conveniently.
>
> Additional minor edits, promised in our discussion with reviewers, can be made at a later date.

---

### Decision · Action_Editor_mPpQ · 2026-06-17

**Recommendation:** Accept with minor revision

**Additional Comments:**

There are two main, closely related problems: (1) correctness and (2) reader friendliness. The current version cannot be read without considerable effort on the part of the reader because the proof style is highly laconic. I acknowledge that in some fields within mathematics and physics, that is acceptable. However, there is a tradeoff: a terse style with short justifications is only acceptable under the condition of absolute 100 percent correctness from the very first submission: a reader is only willing to spend a large amount of time trying to make sense of authors sentences one by one if they are confident that they will eventually make a lot of sense. Once that trust has been broken, there is no choice but to return to a more complete and rigorous style as is customary in machine learning venues.

On correctness:

In their final justification, Reviewer yrCr recommends rejection partially on the basis that they are not confident in the correctness of the proof of Proposition 3 because *"Furthermore, it is easy to check that $H(B_k+X)$
 is maximised where X is uniformly distributed [...]" (top of page 20). Why does
$B_k$ not influence the entropy maximisation too? As a consequence, I can not confirm that the amended proof of Proposition 3 is fully valid."*


Although I agree with the authors that the statement is correct, I do not find the statement trivial at all and I think it requires a separate proof.




On reader friendliness+ some of other comments for the resubmission (these are non exhaustive).


1. The experiments are not really delineated, they are discussed within the theory sections.

2. Although I side with the authors on the issue of the relevance of the topic, it would be interesting to evaluate the proposed "generalized matching pursuit" as a decoder in an applied machine learning context. For instance, if you train a simple sparse autoencoder for on a recommender systems dataset and replace the decoder by generalized matching pursuit, what happens to the performance? Is there a training regime where the use of the algorithm can actually improve performance?

3. "remark 1" reads a bit off in the current version. It appears this is actually making a provable statement about threshold decoding with threshold 1/2. In that case, it should be stated properly and proved.

4. Even "well-known"statements such as the first inequality in Section B should come with a precise citation.

5. Section A isn't really much clearer than its main paper analogue (Section 2.1). If $\lambda_i$ is the maximum likelihood estimator, it should be stated in section 2.1 rather than section A, and Section A should contain more precise computational details. "A routine calculation" feels handwavy.

6. In typical machine learning publications, it is good form to explain each algorithm in detail even if it is provided in the form of an "Algorithm" as your Algorithm 3. One thing I find particularly problematic is the fact that Algorithm 3, which is the only algorithmic contribution of the paper, is described as "a simplification of generalized orthogonal matching pursuit", without explaining the difference with the latter.


7. The logical flow in page 21 is ultimately correct, but takes a lot of effort to take in. It would be clearer to state "indeed, assuming the above claim  holds, we could then set $k'=k-1$.... and deduce that..." and then "thus, assuming claim 1 is valid, we conclude overall that....".

8. Although in this particular case, the authors are correct, it would still be better form to spell out arguments such as the one made on the first line of page 20 more accurately ("the entropy of... is unaffected by applying a rescaling"): although trivial, I think it would follow common practice to explicitly define the rescaled variable $\pi_i(Y)'=d\pi_i(Y)$ and the same for $X$.




Overall, although the reviewers state the novelty is incremental in the final form of the manuscript, I think this work is definitely interesting enough to pass the bar at TMLR, but it must be written clearly and correctly enough to satisfy TMLR's correctness and clarity requirements.

**I am really on the fence about this paper**. Although rejection is perfectly defensible, after careful deliberation, I am tentatively recommending "minor revisions" solely because that is the only option that doesn't trigger a very lengthy new round of reviews (with the randomness associated with a new batch of reviewers). However, **this should not be considered a guarantee that the paper will be accepted**: the required revisions are really major, and this is a way to give the authors one last chance to rewrite the paper and avoid considerable delays. As a guideline, I estimate the authors will need at least 2-3 weeks of full time work trying to fix the paper. If the revised version still looks too vague to me, I will have to reject it and send it through the reviewing process again.

**Audience:**

Yes

**Audience Explanation:**

The paper is concerned with the sample complexity of sparse code recovery with Rademacher dictionaries with one-step methods such as top-k, threshold methods and the newly introduced "generalized matching pursuit" algorithm. The authors make a significant effort to frame the work as relevant to sparse autoencoders. Whilst the reviewers are sceptical of the analogy, I think the topic of sparse recovery is certainly well within scope for TMLR.

**Claims And Evidence:**

No

**Claims Explanation:**

Although the authors appear generally competent and "turned out to be right" in *some* instances of inconsistencies raised by the reviewers (I agree that it is reasonably obvious that a normalized sum of uniform random variables on the sphere is again uniformly distributed on the sphere, for instance), the paper is still far from reader friendly and neither the reviewers nor I can fully vouch for correctness.

By the authors' own admission, both propositions 3 and 4 contained serious problems in their original form and have been **completely rewritten** in the new submission. These constitute the **main claims** of the paper.

The original manuscript made claims about both Rademacher and spherical dictionaries, but due to errors, the authors fell back to only the former. The experiments only cover the latter (spherical dictionaries) and are not discussed separately. To the best of my understanding, the only reviewer who has attempted to read the updated proofs of propositions 3 and 4 claims they may still be incorrect and recommends rejection. Although the reviewer hasn't uncovered an actual error there, I do find the updated proof rushed and difficult to read.

---

> ### Author Response · Authors · 2026-07-17
>
> In the camera ready version we have done our best to resolve these issues. Our updates include:
>
> - Careful clarifications on many aspects of the proofs in appendix, which should now be easier to read. In particular we give a simple proof of the statement on entropy maximization that was called out by yrCr.
> - Clarifications in the appendix on matched filters, and a new appendix detailing the derivation of the maximum a posteriori thresholding level.
> - A short discussion of the gMP algorithm and how it compares to other methods. (We have changed the acronym from GMP to gMP, to match with gOMP.)
> - A separate section delineating the experimental results.
> - A comparison with compressive sensing results, as discussed with YDG8.
>
> In particular we hope to have addressed all the issues on reader friendliness, except (2). While it would be interesting to see how our findings can inform other applications, including recommender systems, we feel that this paper can stand as a mostly theoretical work. However, on the subject of numerical experiments, we have now also performed experiments on Rademacher dictionaries, which are now presented in Figures 3 and 4. Experiments on spherical dictionaries are moved to Appendix G. Our results on both families of dictionary are almost identical.
>
> Finally, we'd like to clarify that, while the proof of Proposition 3 did regrettably have an error in the original version, we're not aware of any error related Proposition 4. We rewrote the proof only to slightly extend the proposition statement; our previous result only applied to $\tau = (1 + \sqrt \eta)^{-1}.$